# Whole-genome sequencing for an enhanced understanding of genetic variation among South Africans

Ananyo Choudhury[1], Michèle Ramsay[1,2], Scott Hazelhurst[1,3], Shaun Aron[1], Soraya Bardien[4], Gerrit Botha[5], Emile R. Chimusa[6], Alan Christoffels[7], Junaid Gamieldien [7], Mahjoubeh J. Sefid-Dashti[7], Fourie Joubert[8], Ayton Meintjes[5], Nicola Mulder[5], Raj Ramesar[6], Jasper Rees[9], Kathrine Scholtz[10], Dhriti Sengupta[1], Himla Soodyall[2,11], Philip Venter[12], Louise Warnich[13] & Michael S. Pepper[14]

The Southern African Human Genome Programme is a national initiative that aspires to unlock the unique genetic character of southern African populations for a better understanding of human genetic diversity. In this pilot study the Southern African Human Genome Programme characterizes the genomes of 24 individuals (8 Coloured and 16 black southeastern Bantu-speakers) using deep whole-genome sequencing. A total of ~16 million unique variants are identified. Despite the shallow time depth since divergence between the two main southeastern Bantu-speaking groups (Nguni and Sotho-Tswana), principal component analysis and structure analysis reveal significant ($p < 10^{-6}$) differentiation, and $F_{ST}$ analysis identifies regions with high divergence. The Coloured individuals show evidence of varying proportions of admixture with Khoesan, Bantu-speakers, Europeans, and populations from the Indian sub-continent. Whole-genome sequencing data reveal extensive genomic diversity, increasing our understanding of the complex and region-specific history of African populations and highlighting its potential impact on biomedical research and genetic susceptibility to disease.

[1] Sydney Brenner Institute for Molecular Bioscience, Faculty of Health Sciences, University of the Witwatersrand, Johannesburg 2193, South Africa. [2] Division of Human Genetics, School of Pathology, Faculty of Health Sciences, University of the Witwatersrand, Johannesburg 2000, South Africa. [3] School of Electrical and Information Engineering, University of the Witwatersrand, Johannesburg 2050, South Africa. [4] Division of Molecular Biology and Human Genetics, Faculty of Medicine and Health Sciences, Stellenbosch University, Tygerberg 7505, South Africa. [5] Computational Biology Division, Department of Integrative Biomedical Sciences, IDM, University of Cape Town, Cape Town 7925, South Africa. [6] Division of Human Genetics, Department of Pathology, IDM, Faculty of Health Sciences, University of Cape Town, Cape Town 7925, South Africa. [7] South African MRC Bioinformatics Unit, South African National Bioinformatics Institute, University of the Western Cape, Bellville 7925, South Africa. [8] Department of Biochemistry and Genomics Research Institute, Centre for Bioinformatics and Computational Biology, University of Pretoria, Pretoria 0083, South Africa. [9] Agricultural Research Council, Pretoria 0184, South Africa. [10] Department of Preclinical Sciences, School of Health Care Sciences, Faculty of Health Sciences, University of Limpopo, Mankweng 0727, South Africa. [11] National Health Laboratory Service, School of Pathology, Faculty of Health Sciences, University of the Witwatersrand, Johannesburg 2000, South Africa. [12] Department of Medical Sciences, School of Health Sciences, Faculty of Health Sciences, University of Limpopo, Mankweng 0727, South Africa. [13] Department of Genetics, Stellenbosch University, Stellenbosch 7600, South Africa. [14] Institute for Cellular and Molecular Medicine, Department of Immunology, Faculty of Health Sciences, University of Pretoria, Pretoria 0084, South Africa. Ananyo Choudhury and Michèle Ramsay contributed equally to this work. Correspondence and requests for materials should be addressed to M.R. (email: Michele.ramsay@wits.ac.za) or to M.S.P. (email: michael.pepper@up.ac.za)

African populations harbor the greatest genetic diversity[1–5] and have the highest per capita health burden (WHO), yet they are rarely included in large genome studies of disease association[6–8]. The complex history of the people of sub-Saharan Africa is reflected in the diversity of extant populations and recent migrations that have led to extensive regional admixture[9–11]. This diversity provides both a challenge and an opportunity for biomedical research and the hope that Africans will one day benefit from genomic medicine.

Present day South Africans include a major ethnolinguistic group of black southeastern Bantu-speakers (79.2% of the population), an admixed population (including European, Southeast Asian, South Asian, Bantu-speaking African, and hunter gatherer ancestries) referred to as Coloured (COL)[12–14] (8.9%), whites of European origin (8.9%), an Indian population originating from the Indian sub-continent (2.5%), and a small proportion of additional ethnolinguistic affiliations not broadly covered in the aforementioned (http://www.statssa.gov.za/). The focus of this pilot study from the Southern African Human Genome Programme (SAHGP) is on the southeastern Bantu-speaker and COL populations.

Archeological evidence suggests that the migration of groups of Bantu-speaking agro-pastoralists into southern Africa was initiated about 2000 years ago[15–17]. It further supports two different migration paths, one in the east and one in the west of Africa, giving rise to southeastern Bantu-speaker (SEBs) and southwestern Bantu-speakers (SWB)[15, 18]. Migration of SEB is estimated to have occurred in multiple distinct waves (in the early, middle, and late iron age) along the eastern coast[19–23]. The patterns of distribution of artifacts and rock art from different iron-age sites indicate the complex nature of the interactions between the Bantu-speaking immigrants and the Khoesan (KS) inhabitants[24]. These involved long phases of coexistence, trade, assimilation of hunter–gatherer peoples into agro-pastoralist communities, and in some cases the displacement of KS groups[25–28]. Such interactions have not only involved linguistic and cultural exchange but also admixture at the genetic level[29, 30]. It can be postulated that the migration of each Bantu-speaking group into a new territory likely involved an independent set of interactions and admixture events with the resident agro-pastoralist Bantu-speaker and hunter–gatherer populations.

These migrations and interactions have led to the formation of ethnolinguistic divisions within the SEB of present day South Africa, of which the two major groups are the Nguni-speakers and Sotho-Tswana-speakers who are estimated to have diverged geographically over the past 500 years or so[21]. The Nguni-speakers expanded to occupy the coastal areas extending down the east coast of South Africa, whereas Sotho-Tswana-speakers expanded across the highland plateau between the eastern escarpment and the more arid regions in the west[21]. Although the details of the arrival of these populations are unclear, it is proposed that the Nguni- and Sotho-Tswana-speakers or their antecedents migrated to southern Africa and started occupying vast territories by the fifteenth century[21, 22, 31]. Some of the boundaries between these populations have, however, been obscured by more recent migrations, conquests, admixture, and, in some cases, rapid language adaptations, especially over the last two centuries[32]. This makes the consideration of geography and language important when assessing the divergence of these groups and begs the question as to whether genetic studies would be sufficiently powered to detect population differences.

Southern African populations have recently been investigated using a number of genomic approaches including genotyping array and, more rarely, whole-genome sequencing (WGS) technologies[9, 13, 29, 30, 33–38]. However, the focus of most of these studies has been to analyze the genomic diversity among hunter–gatherers and the extent of their admixture in the present day SEB[13, 29, 30, 36, 38]. An early study, based on mtDNA, Y-chromosome, and a limited number of autosomal markers, suggested that the ethnolinguistic divisions between the major SEB groups were reflected by observed genetic divergence, although the clustering was not consistent for the three data types[39]. More recent genome-scale studies have not replicated the substructure within the SEB[9, 38] and in some cases the authors concluded that the SEB is genetically a relatively homogenous group. This assumption needs more thorough investigation.

Over the past century there has been extensive urbanization of SEB in South Africa and the migration to economic hubs has resulted in a confluence of multiple ethnolinguistically diverse groups (http://www.statssa.gov.za/). When recruiting study participants from urban settings, the ethnolinguistic boundaries become blurred and the distinctions are no longer evident. We have therefore purposely recruited the SEB for this study from rural and semi-urban regions where we anticipated little or no ethnolinguistic admixture.

The arrival and settlement of Europeans during the last 500 years is an important migration that has influenced the peopling of southern Africa[12, 40]. Slave trade into the Western Cape from the 1600s also brought South Asian and Southeast Asian people to South Africa[12]. The interactions between these populations, Bantu-speakers and KS, have given rise to complex admixed population, one of which is the five-way admixed Cape COL population[12]. The recent and complex admixture patterns of the COL populations from different geographic regions and religious affiliations have been investigated in many different studies[12–14, 30, 40, 41]. They confirmed the presence of at least five ancestral populations and demonstrate significantly different ancestry proportions among individuals sampled from different regions of South Africa[13, 40, 42]. These studies were all based on SNP-array data and, to date, no WGS data have been published from COL individuals. Largely due to unavailability of data from appropriate ancestral populations, representation of populations from Southeast and South Asia may, in some instances, have biased the estimate of ancestry proportions.

The focus of the SAHGP pilot study is to provide a WGS-based, unbiased estimate of genetic variation in the region and to study the genetic differences between some of the major ethnolinguistic groups in the country. The study included 24 ethnically self-identified individuals comprising 16 SEBs (seven Sotho-Tswana- and nine Nguni-speakers) and 8 COL individuals. The first major aim was to study possible correlations between language groups and genetic clustering. The results suggest that, at least for individuals sampled on the basis of both language and geographic location, there is a discernable genetic separation between the two major SEB ethnolinguistic groups. The second major aim was to investigate the ancestral composition of the COL individuals based on novel WGS data and a comprehensive assortment of potential ancestral populations. As a result of the inclusion of additional representative populations our study demonstrates a much stronger South Asian ancestry in the COL when compared to previous studies. We document significant novel SNV discovery from the 24 WGS and highlight the potential implications for disease susceptibility in Africans.

## Results

**Description of variants discovered.** This study used deep WGS (~50×; Supplementary Table 1) data to provide an unbiased assessment of genomic variation in 24 apparently healthy South African male individuals. In an attempt to capture a spectrum of diversity in under-represented populations we included eight

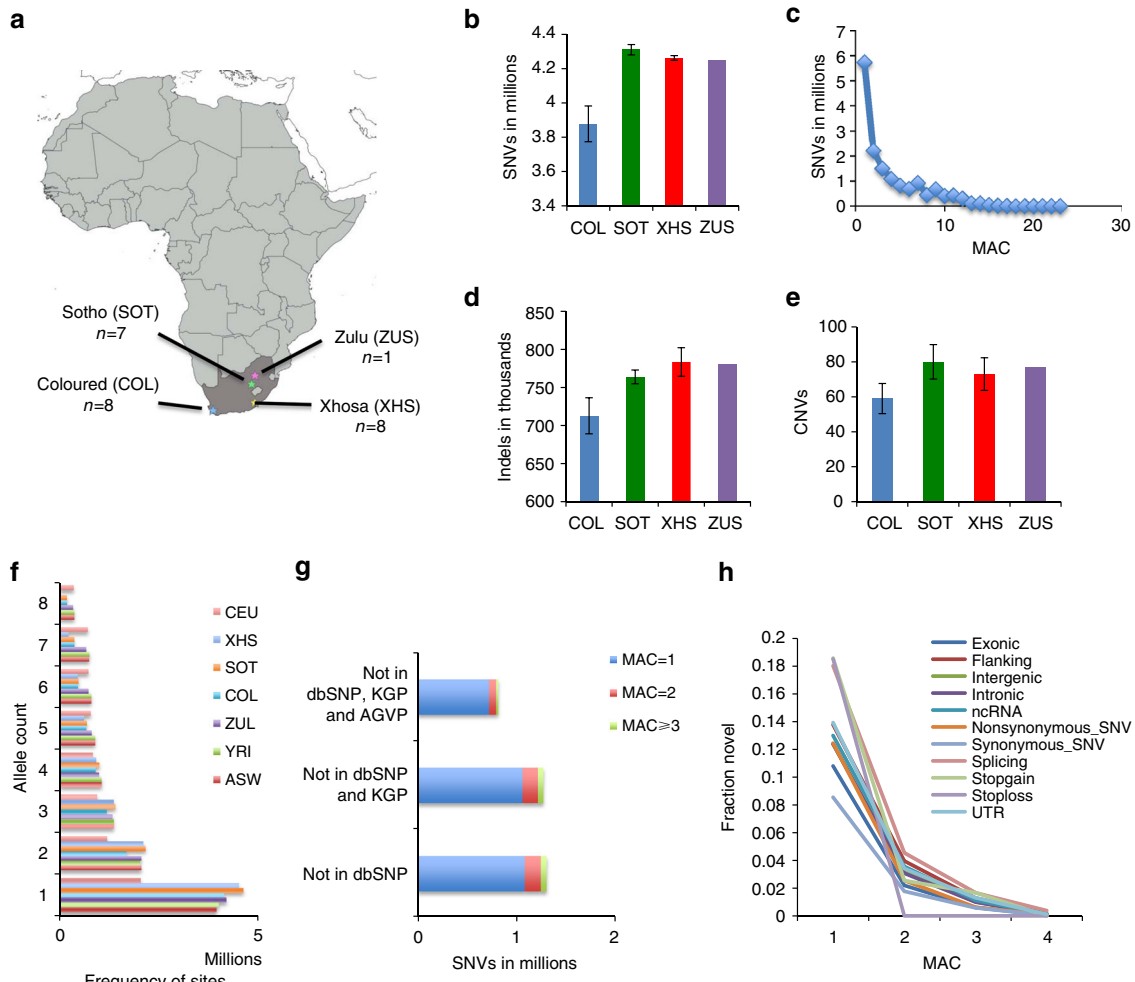

**Fig. 1** SAHGP participants and genetic variants detected by high-coverage whole-genome sequencing in 24 South Africans. **a** Current geographic location of the participants: Coloured (COL) is a group of mixed ancestry individuals from the Western Cape with historically predominantly Malay, Khoesan, European, Indian sub-continent, and black African admixture. Sotho-speakers (SOT) were from the small rural town of Ventersburg in the Free State Province and represent the Sotho-Tswana language speakers. Xhosa-speakers (XHS) were from a clinic in Port Elizabeth in the Eastern Cape in a region with relatively low recent in migration. The ZUS individual was a Zulu-speaker from Soweto (XHS and ZUS represent the Nguni language speakers). In South Africa, the two main linguistic subgroups among southeastern Bantu speakers are the Nguni and Sotho-Tswana. The map was generated using SimpleMappr (http://www.simplemappr.net/). **b** Average number of SNVs detected per individual from the three groups showing that the COL individuals had fewer non-reference alleles than the Bantu speakers. **c** Minor allele count (MAC) distribution of SNVs. **d** Average number of indels detected per individual. **e** Average number of CNV detected per individual. **f** Site frequency spectrum in the three SAHGP populations in comparison to equal-sized samples drawn from Utah residents (CEPH) with Northern and Western European Ancestry (CEU), Zulu from South Arica (ZUL), Yoruba from Ibadan, Nigeria (YRI), and Americans of Africa ancestry from south west USA (ASW). Eight samples were randomly drawn from each of the populations, and the values shown are the average of five random sets. **g** Novel SNVs discovered in the study and their MACs shown in different *colors*. The novel SNVs were defined in comparison to the 1000 Genomes Project Phase 3 (KGP), dbSNP142, and the African Genome Variation Project (AGVP) data sets. **h** The relative representation of novel SNVs in each functional class of SNV in the data set

individuals of mixed ancestry from the Western Cape (referred to as COL) and 16 black South African SEB (7 Sotho-speakers from the Free State (SOT), eight Xhosa-speakers (XHS) from the Eastern Cape and 1 Zulu-speaker (ZUS) from Gauteng; Fig. 1a). Single-nucleotide variants (SNVs) were called using three different approaches with high concordance and only SNVs called by all three were used for downstream analyses (Supplementary Table 2 and Supplementary Note 1). Indels and copy number variants (CNVs) were called according to the standard Illumina pipeline. The analysis approach is outlined in Supplementary Fig. 1a. The average number of SNVs, indels, and CNVs was markedly higher in the black South Africans compared to the COL individuals (Fig. 1b–d and Supplementary Tables 2, 3). Across the 24 samples, 16.3 million unique SNVs were identified.

A significant proportion of the SNVs identified were singletons (Fig. 1e). Interestingly, the number of singletons in SOT and XHS was found to be higher in comparison to singletons detected in randomly selected low-coverage African WGS sets of equal size (Fig. 1f, Supplementary Notes 1, 2); however, the observed differences in addition to demographic factors might also reflect the differences in sequencing coverage among the studies[5, 43, 44] (Supplementary Notes 1, 2).

SNVs and indels were annotated according to genic locations using ANNOVAR[45] (Supplementary Tables 4, 5). A total of 3936 unique loss of function (LOF) candidate variants, which included stop gain, stop loss, splice, and frameshift mutations, were observed (Supplementary Fig. 2). The list was pruned by excluding variants observed at a MAF > 0.01 in 1000 Genomes

**Table 1 SNVs showing potential knockout configurations in 16 genes that have two or more LOF mutations in the heterozygous state in the same individual**

| Gene | Position | COL[a] | ZUS[a] | SOT[a] | XHS[a] | Type |
|------|----------|------|------|------|------|------|
| LILRA3 | 19_54803664 | 1 | 0 | 0 | 0 | Stopgain |
|  | 19_54803979 | 1 | 0 | 0 | 0 | Splicing |
| SLC17A9 | 20_61588315 | 1 | 0 | 0 | 0 | Splicing |
|  | 20_61588316 | 1 | 0 | 0 | 0 | Splicing |
| UGT2A3 | 4_69817185 | 0 | 0 | 1 | 0 | Frameshift_deletion |
|  | 4_69796262 | 0 | 0 | 1 | 0 | Splicing |
| AC006486.1 | 19_42747163 | 1 | 0 | 0 | 0 | Frameshift_deletion |
|  | 19_42747179 | 1 | 0 | 0 | 0 | Frameshift_deletion |
| PLSCR2 | 3_146179745 | 0 | 0 | 0 | 1 | Splicing |
|  | 3_146177635 | 0 | 0 | 0 | 1 | Frameshift_deletion |
| ETNPPL | 4_109681449 | 0 | 1 | 0 | 0 | Frameshift_deletion |
|  | 4_109681452 | 0 | 1 | 0 | 0 | Stopgain |
| ZNF816 | 19_53454007 | 0 | 0 | 0 | 1 | Frameshift_deletion |
|  | 19_53454370 | 0 | 0 | 0 | 1 | Stopgain |
| AC026740.1 | 5_668574 | 2 | 0 | 0 | 0 | Frameshift_insertion |
|  | 5_668654 | 2 | 0 | 0 | 0 | Frameshift_deletion |
| AC078925.1 | 12_131514221 | 0 | 0 | 0 | 1 | Frameshift_deletion |
|  | 12_131514761 | 0 | 0 | 1 | 1 | Frameshift_insertion |
| AC078925.1 | 12_131514265 | 0 | 0 | 0 | 1 | Frameshift_insertion |
|  | 12_131514264 | 0 | 0 | 1 | 0 | Frameshift_deletion |
| IGSF22 | 11_18728743 | 0 | 0 | 0 | 1 | Frameshift_deletion |
|  | 11_18727647 | 0 | 0 | 0 | 1 | Frameshift_deletion |
| FNDC3A | 13_49775314 | 0 | 0 | 0 | 1 | Frameshift_deletion |
|  | 13_49775366 | 0 | 0 | 0 | 1 | Splicing |
| AGAP6 | 10_51748681 | 1 | 0 | 1 | 0 | Frameshift_deletion |
|  | 10_51768674 | 1 | 0 | 0 | 1 | Frameshift_deletion |
|  | 10_51748528 | 0 | 0 | 0 | 1 | Frameshift_insertion |
| SORBS3 | 8_22432388 | 1 | 0 | 0 | 1 | Frameshift_deletion |
|  | 8_22432396 | 1 | 0 | 0 | 1 | Stopgain |
| LRRC9 | 14_60448779 | 1 | 0 | 0 | 0 | Splicing |
|  | 14_60474859 | 1 | 0 | 0 | 0 | Stopgain |
| CDHR3 | 7_105668924 | 0 | 0 | 1 | 0 | Splicing |
|  | 7_105641910 | 0 | 0 | 1 | 0 | Stopgain |
| AC008686.1 | 19_13899040 | 0 | 0 | 0 | 1 | Frameshift_deletion |
|  | 19_13899019 | 0 | 0 | 0 | 1 | Splicing |

[a] See Supplementary Table 7 for further detail. The number of individuals tested per group: COL ($n = 8$), SOT ($n = 7$), ZUS ($n = 1$), and XHS ($n = 7$)

Project Phase 3 (KGP)[46] and the African Genome Variation Project (AGVP)[9], resulting in 1703 variants. Their gene locations were determined and 146 genes had at least two LOF variants in the data set, of which 22 genes showed a potential knockout configuration (two heterozygous LOF variants in the same individual) in at least one individual. Six of the genes were excluded because they are listed in the false discovery panel[47] and the LOF variants in the remaining 16 genes are shown in Table 1 and Supplementary Table 6. These genes are not associated with known phenotypes in OMIM (http://www.omim.org/), with the exception of SLC17A9, which has variants segregating with autosomal dominant disseminated superficial actinic porokeratosis-8 in two unrelated Chinese families (http://wwwgenecards.org).

**Novel variants.** Of the 16.3 million unique SNVs identified, 815,404 were detected to be novel (defined as absent from dbSNP142[48], KGP[46], and the AGVP study[9] (Supplementary Table 7)). Novel SNVs were categorized according to minor allele count (MAC), with the largest proportion of variant alleles observed only once (Fig. 1g). The large number of novel variants demonstrates the potential for novel discovery in African populations. The representation of novel SNVs in various functional categories was also studied and is summarized in Fig. 1h (Supplementary Note 3). The distribution of novel SNVs across the genome is shown in Supplementary Fig. 3 and Supplementary

Table 8 and highlights regions of high density and potential interest (Supplementary Note 3). Regions with high overall SNV density differences between black South Africans and other African populations were identified (Supplementary Note 4). Several of these regions were found to be associated with protein-coding genes (Supplementary Table 9). Local ancestry analysis of these regions may reveal hotspots for mutational activity or enrichment of haplotype blocks from specific ancestral populations (e.g., the KS).

**Population structure and admixture.** Recent historical events including geographic isolation, cultural practices, political conflict, colonization, and extensive admixture have shaped the genetic diversity among populations of southern Africa[11, 30, 49]. Comparative studies for population structure and admixture were done using SNP-array data available in the public domain[9, 30, 46] (see Supplementary Table 10 for the list of populations used). Fig. 2a shows global data and Fig. 2b focuses on Africa.

Principal component analysis (PCA) showed that the COL individuals form a dispersed cluster linked to African and non-African populations including European, South-Asian (Indian sub-continent), and Austronesian populations[50], confirming their parental contributions as reported in historical accounts (Fig. 2, Supplementary Fig. 4, and Supplementary Note 5). The analysis of ancestry proportions based on novel proxy populations provided an indication of substantive admixture from the Indian

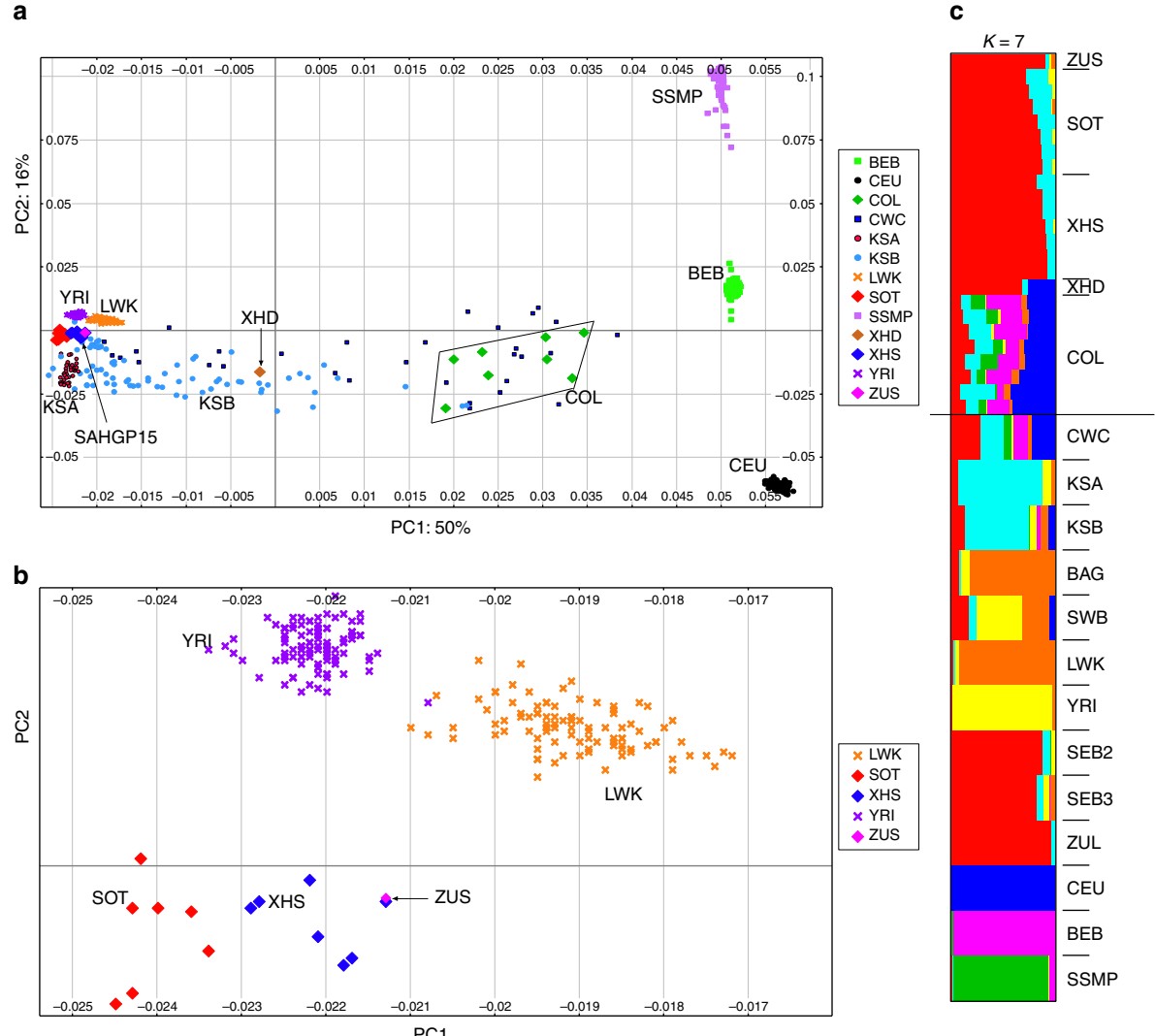

**Fig. 2** Genetic differentiation of Sotho and Xhosa-speakers demonstrated by principal component analysis (PCA) and ADMIXTURE analysis. PCA was performed using the Illumina 2.5 M array data for comparative purposes. Overall, 197,279 SNVs were used in the analysis. **a** Comparison of South Africans relative to selected world populations (also in Supplementary Fig. 4; PC1, 2, and 3 explain 50, 16, and 14% of structure variation, respectively). **b** This shows the same data and analysis but zooming-in on selected populations for clarity. The Sotho (SOT) and Xhosa (XHS) show distinctive clustering, and thereby suggest significant genetic differentiation. **c** ADMIXTURE analysis was done with a selection of world populations. A summarized result for $K = 7$ is shown here (more details in the Supplementary Fig. 7a, b). Above the *horizontal line* each individual in the SAHGP sample is shown, one *row* per person; below the *line* we show the average ancestral composition for all members of that group. The populations included from the present study are Sotho (SOT), Xhosa (XHS), Zulu from Soweto (ZUS), Coloured (COL), and the admixed Xhosa individual from South Africa (XHD). Additional populations used in this analysis are Baganda from Uganda (BAG), Bengali from Bangladesh (BEB), Utah Residents (CEPH) with Northern and Western European Ancestry (CEU), COL from Wellington (CWC), Northern and Central Khoesan including Ju/'Ōhoansi, Glui, and Gllana and !Xuun (KSA), Southern Khoesan including Khwe, Karretjie, Nama and ≠Khomani (KSB), Luhya in Webuye, Kenya (LWK), South Eastern Bantu speakers from Schlebusch et al. 2012 (SEB2); Black South Africans from Soweto based on May et al. 2013 (SEB3), Malay from Singapore (SSMP); southwestern Bantu-speakers (SWB); Yoruba in Ibadan, Nigeria (YRI); Zulu from South Africa (ZUL). Further information on these populations is available in Supplementary Table 10

sub-continent along with contributions from the European, KS, SEB, and the Austronesians (Malay; Fig. 2c, Supplementary Table 10, Supplementary Fig. 5, and Supplementary Note 6). However, it needs to be noted that the admixture among groups of COL individuals is known to differ significantly along religious lines and geographic dispersal[51, 52].

Despite the recent linguistic and geographic divergence between the XHS and SOT groups, genetic data using PCA showed them to be significantly different ($p < 10^{-6}$; Fig. 2 and Supplementary Fig. 6). The genetic structure between the two groups was also reflected in the structure analysis (Fig. 2c and Supplementary Fig. 7).

One individual who self-identified as XHS was found to have recent non-African admixture of European origin (he is identified as XHD in Fig. 2a), leaving 15 individuals in the SEB group. The ZUS individual did not seem to cluster with the AGV Zulu participants (see Supplementary Fig. 6e)[9]. The ZUS individual was recruited in Soweto, Johannesburg, which is a cosmopolitan area having attracted migrants from across southern Africa, with different ethnic backgrounds. Soweto has a complex history, including people who were forcibly relocated there under apartheid legislation from other urbanized areas in the 1950s[53]. Thus, Soweto has an effective 120-year history of people from different backgrounds living together in an urbanized setting.

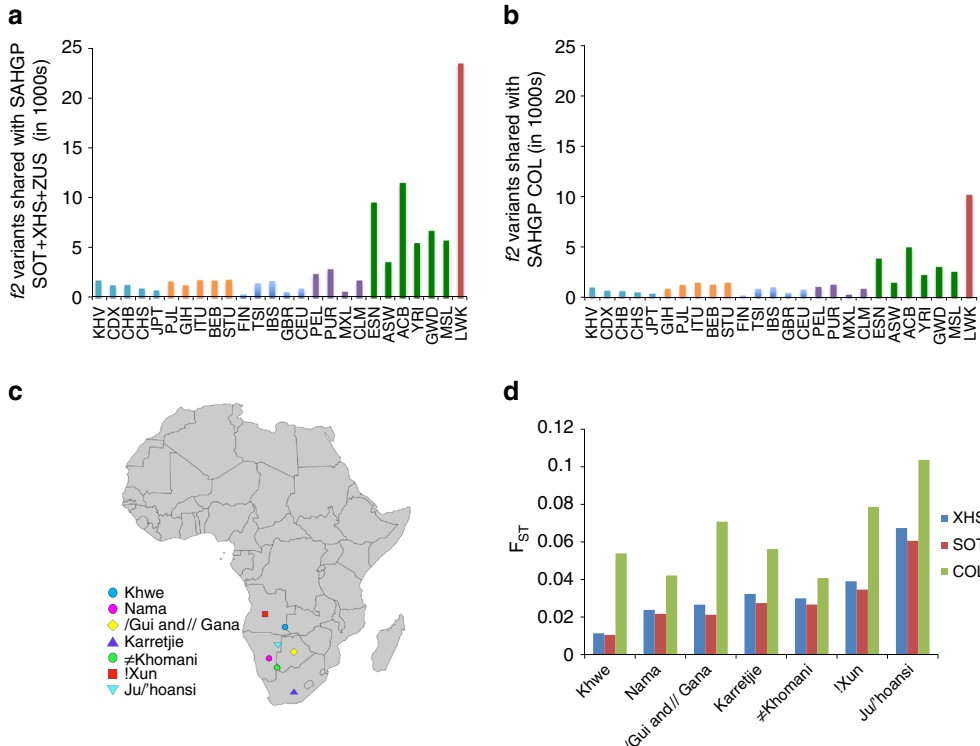

**Fig. 3** Relatedness of the South Africans to other populations estimated on the basis of variant sharing and allelic differentiation. Comparison of variants found by whole-genome sequencing in the **a** Southeastern Bantu-speakers (SEB) and **b** Coloured (COL) from this study with the 1000 Genomes Project (KGP) populations using the $f_2$ estimate. Only the occurrence of $f_2$ variants shared between the South African populations and the KGP populations are shown. The analysis suggests a more recent historical connection between southern and East African Bantu speakers. **c** Map highlighting the geographic region of southern African hunter–gatherer groups used for comparison in the study. The map was generated using SimpleMappr (http://www. simplemappr.net/). **d** $F_{ST}$ values showing comparison between Sotho-speakers (SOT), Xhosa-speakers (XHS), COL populations, and previously studied southern African hunter–gatherer populations. The South African Bantu-speaking groups (SOT and XHS) were found to be closest to the Khwe, whereas the COL was found to be closest to the Nama and ≠Khomani. The y-axis shows the average $F_{ST}$ value for each comparison. KGP populations used in this analysis along with Color codes are: East Asian (cyan)—Kinh in Ho Chi Minh City, Vietnam (KHV); Chinese Dai in Xishuangbanna, China (CDX); Han Chinese in Bejing, China (CHB): Southern Han Chinese (CHS); Japanese in Tokyo, Japan (JPT), South Asian (orange)—Punjabi from Lahore, Pakistan (PJL); Gujarati Indian from Houston, Texas (GIH); Indian Telugu from the UK; Bengali from Bangladesh (BEB); Sri Lankan Tamil from the UK (STU), European (blue)—Finnish in Finland (FIN); Toscani in Italia (TSI); Iberian Population in Spain (IBS); British in England and Scotland (GBR); Utah Residents (CEPH) with Northern and Western European Ancestry (CEU), Admixed Americans (purple)—Peruvians from Lima, Peru (PEL); Puerto Ricans from Puerto Rico (PUR); Mexican Ancestry from Los Angeles USA (MXL); Colombians from Medellin, Colombia (CLM), West, Central-West African and African descent (green)— Esan in Nigeria (ESN); Americans of African Ancestry in SW USA (ASW); African Caribbeans in Barbados (ACB)—Yoruba in Ibadan, Nigeria (YRI); Gambian in Western Divisions in the Gambia (GWD); Mende in Sierra Leone (MSL); and East African (red)—Luhya in Webuye, Kenya (LWK)

Although there is only one ZUS individual in the study and so comment risks being anecdotal, the fact that the ZUS individual does not cluster in our sample with the ZUL individuals (from the AGVP data set) indicates that care needs to be taken in interpreting ethnic origin when recruiting from urbanized areas in African countries. Language and self-identity may not be good markers for genetic background.

**Regions of genomic differentiation between Sotho and Xhosa.** To investigate the differentiation of SOT and XHS further, we studied the distribution of average fixation index ($F_{ST}$) scores (in 25 kb windows) across the genome (Supplementary Note 7). The analysis of regional $F_{ST}$ was able to identify genomic regions with high divergence between the two groups (Supplementary Fig. 8 and Supplementary Table 11). Although a large proportion of the high $F_{ST}$ windows was found to occur in the intergenic regions and pseudogenes, some of the windows were found to include the olfactory receptor genes OR4S2 and OR4C6, and other genes like SEMA4F, EREG, PLN, and PTF1A (Supplementary Fig. 8 and Supplementary Table 11). The potential biological roles of these genes were inferred using the Genecards database

(http://wwwgenecards.org). The gene SEMA4F encodes a trans-membrane class IV semaphorin family protein, which plays a role in neural development. This gene has been suggested to be involved in neurogenesis related to prostate cancer, the development of neurofibromas, and breast cancer tumorigenesis. In addition to cancers, the SEMA4F gene has also been suggested to be involved in pulmonary tuberculosis and dyslexia. The EREG (epiregulin) gene can function as a ligand of EGFR (epidermal growth factor receptor), as well as a ligand of most members of the ERBB (v-erb-b2 oncogene homolog) family of tyrosine-kinase receptors and is known to be associated to diseases like colorectal cancer, and hypopharynx cancer. Similarly, the PTF1A gene is a transcription factor involved in pancreatic development. Diseases associated with PTF1A include pancreatic cancer and cerebellar agenesis. In the absence of data for variation in phenotype or disease incidence or prevalence in these groups, it is not possible to infer whether the highlighted genes have any medical or evolutionary significance for these populations. Nevertheless, the genetic differences in these regions might flag some of these diseases/traits for epidemiological investigation among the southern African populations.

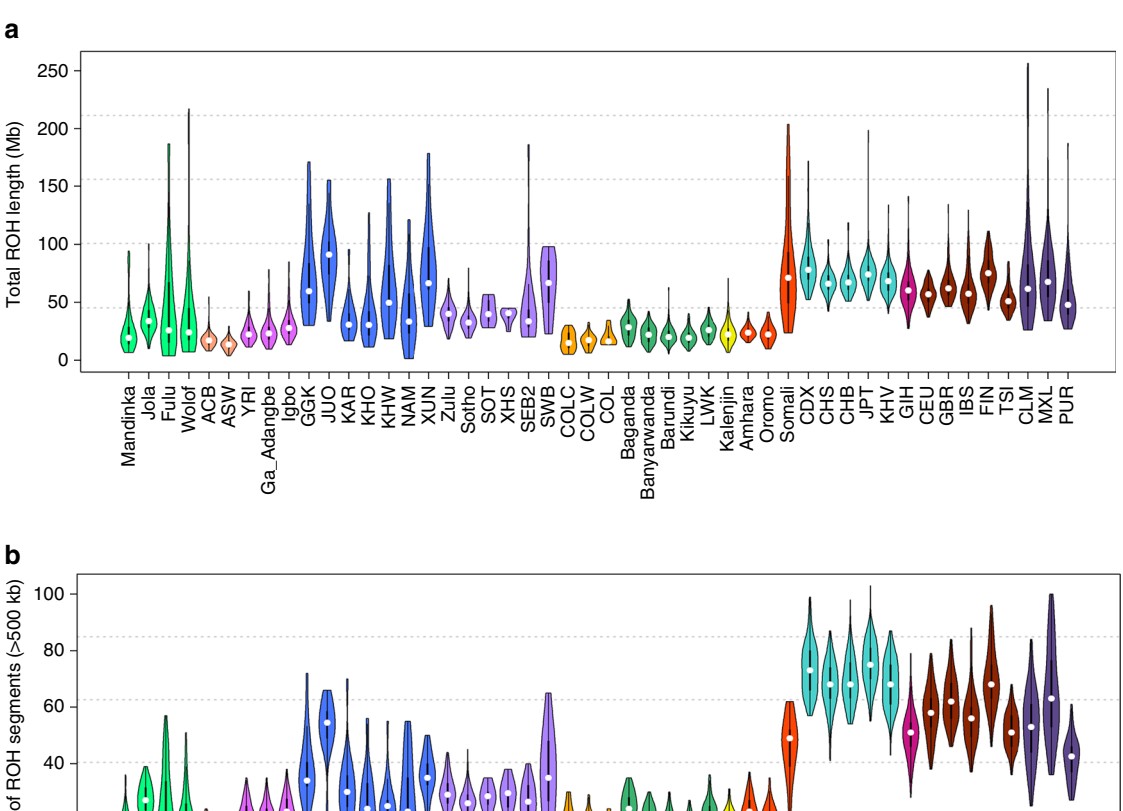

**Fig. 4** South African Bantu speakers show relatively higher proportions of runs of homozygosity segments compared to most Bantu speakers. **a** Total runs of homozygosity (ROH) length in Mb (median per population) and **b** the number of ROH segments in various African and non-African populations. Violin plots show median (*white dot*) and range with width indicating frequency. Each *color* corresponds to a super-population group (Supplementary Table 11). Populations used in this analysis include samples from: West Africa (shown in *light green*)—Mandinka; Jola; Fula; Wolof, Admixed Africans (shown in *light orange*)—African Caribbeans in Barbados (ACB); Americans of African Ancestry in SW USA (ASW), Central-West African (shown in *magenta*): Yoruba in Ibadan, Nigeria (YRI); Ga_Adangbe; Igbo, Khoesan (shown in *blue*)—Glui, Gllana, and Kagalgadi (GGK); Ju/̃Ohoansi (JUO); Karretjie (KAR); ≠Khomani (KHO); Khwe (KHW); Nama (NAM); and !Xuun (XUN), South African Niger-Congo-speakers (shown in *light purple*)—Zulu; Sotho; Sotho (SOT), Xhosa (XHS); South Eastern Bantu speakers (SEB2); South Western Bantu speakers (SWB). Admixed South Africans (shown in *ochre*), Coloured from Colesberg (COLC); COL from Wellington (COLW), Eastern African Niger-Congo-speakers (shown in *deep green*)—Baganda; Banyarwanda; Burundi; Kikuyu; Luhya in Webuye, Kenya (LWK), Nilo-Saharan-speakers (shown in *yellow*)—Kalenjin, Afro-Asiatic-speakers (shown in *red*)—Amhara; Oromo; Somali, East Asia (shown in *sea green*)—Chinese Dai in Xishuangbanna, China (CDX); Southern Han Chinese (CHS); Han Chinese in Bejing, China (CHB); Japanese in Tokyo, Japan (JPT); Kinh in Ho Chi Minh City, Vietnam (KHV), South Asian (shown in *deep magenta*)—Gujarati Indian from Houston, Texas (GIH), European (shown in *brown*)—Utah Residents (CEPH) with Northern and Western European Ancestry (CEU); British in England and Scotland (GBR); Iberian Population in Spain (IBS); Finnish in Finland (FIN); Toscani in Italia (TSI); and Admixed Americans (shown in *deep purple*)—Colombians from Medellin, Colombia (CLM); Mexican Ancestry from Los Angeles USA (MXL); and Puerto Ricans from Puerto Rico (PUR). Further information on these populations is available in Supplementary Table 12

**Affinities of populations based on rare variant sharing**. To further understand the genetic affinities of the South Africans, we performed $f_2$ analysis using WGS data from the KGP[46]. If a variant occurred only twice in the merged SAHGP-KGP data set, such that one copy was observed in the SEB, the KGP[46] population with the other copy was noted (Supplementary Note 8). The SEB included 7 SOT, 7 XHS, and the ZUS individual. The frequency of $f_2$ variants shared with SEB by various KGP[46] populations is summarized in Fig. 3a, demonstrating that the majority of $f_2$ variants was shared with the Luhya from Kenya. There was also significant but less sharing with other African populations (especially ACB and ESN) and

low sharing with non-African populations. This trend was also observed when the analysis was extended to examining SNVs, irrespective of MAC, shared between the SEB and a single KGP[46] population (Supplementary Fig. 9). The COL also showed the same pattern of $f_2$ variant sharing with the KGP[46] populations (Fig. 3b). The higher sharing of $f_2$ variants between South African populations with the East African Niger-Congo-speakers compared to West African Niger-Congo-speakers is consistent with the historical accounts of Bantu migration[49, 54]. The distribution of continent-specific variants also demonstrated a similar pattern (Supplementary Fig. 9 and Supplementary Note 8).

**Table 2 Mitochondrial and Y chromosome haplogroup distribution in the 24 SAHGP individuals**

| Sample ID | mtDNA haplogroup | Probable origin | Y Haplogroup | Probable origin |
|---|---|---|---|---|
| COL_A | L0d2a1 | KS | J1b2b3a1b [J-YSC76] | Middle-Eastern |
| COL_B | L0d1b2b1b1 | KS | R1a1a1a1 [R-L664] | European |
| COL_C | L3d1a1a | Bantu-speakers, African Americans | R1b1a2a1a2b2b1a1* [R-Z8*] | European |
| COL_D | M3a1 + 204 | Indians, Chinese, Tibetans | J1b3 [J-Z1828] | Central Europe to Central Asia |
| COL_E | H2a2a1 | Europe, North Africa, Middle East | Ambiguous: [E-P9.2]/[A-P71] | African |
| COL_F | L0d1b2b1b1 | KS | I1a1c1 [I-P109] | Eastern Europe |
| COL_G | L0d2a1a | KS | I1a3* [I-Z63*] | Northern Europe |
| COL_H | L0d2a1 | KS | E1b1a1a1g1* [E-U209*] | African |
| ZUS | L2a1a2a1 | Bantu-speakers, African Americans | E2b1a* [E-M200*] | African, possible KS |
| SOT_A | L0d2c1b | KS | E1b1a1a1g1a1 [E-U181] | African, possibly central |
| SOT_B | L0d3b1 | KS | B2a1a2a2a [B-P50] | African |
| SOT_C | L2a1f | Bantu speakers, African Americans | E1b1a1a1f1a1d [E-CTS8030] | African |
| SOT_D | L0d2a1a | KS | Ambiguous: E-P9.2/A-M51* | African |
| SOT_E | L0d2a1a | KS | A3b1c [A-V306] | African |
| SOT_F | L3f1b4a1 | Yoruba, Fulbe, African Americans | E1b1a1a1f1a1d [E-CTS8030] | African |
| SOT_G | L0d2a1 | KS | Ambiguous: E-P9.2/A-M51* | African |
| XHS_A | L0d1b 2b2b1 | KS | E1b1a1a1f1a1* [E-U174*] | African |
| XHS_B | L3e1b2 | Bantu speakers, African Americans | E1b1a1a1g1a2 [E-Z1725] | African, possible KS |
| XHS_C | L0d1a1b | KS | E1b1a1a1f1a1* [E-U174*] | African |
| XHS_D | L0a2a2a1 | Bantu speakers, Mbuti, Biaka | Ambiguous: [E-P9.2]/[B-P6] | African |
| XHS_E | L0d2a1a | KS | E2b1a* [E-M200*] | African |
| XHS_F | L0d2a1a | KS | E1b1a1a1g1* [E-U209*] | African |
| XHS_G | L0d1a1c | KS | E1b1a1a1f1a1* [E-U174*] | African |
| XHD | L0d1c1a1a | KS | J2a1b2a1* [J-M92*] | Mediterranean/Levant/Europe/Central Asia |

**Characterization of Khoesan affinities**. An important characteristic that distinguishes the SEB from other Africans is the relative proportion of KS admixture[9, 29, 49, 55–57]. Genetic distance between the KS and the SAHGP populations was estimated using $F_{ST}$. As expected, the SOT and XHS showed closer affinity with the Zulu and other SEB from South Africa compared to the other Africans (Supplementary Fig. 10 and Supplementary Note 9). When comparing the COL, SOT, and XHS to different KS groups, the COL showed greater genetic distances than either the XHS or SOT. Relative to the XHS, the SOT consistently showed smaller $F_{ST}$ values, demonstrating that the KS had contributed to the gene pools of the SOT, XHS, and COL populations to varying degrees (Fig. 3c, d). The genetic distances of the COL reflected the geographic proximity of current day KS population dispersal, suggesting that this provided the impetus for admixture. The genetic distance for SOT and XHS, however, showed a more complex pattern of variation with geography that could be due to variation in levels of Bantu-speaking admixture in the KS populations (e.g., the Khwe)[15]. To reduce bias on the $F_{ST}$ estimates introduced due to admixture, we used PCAdmix to identify and mask genomic regions with non-Niger-Congo (NC) ancestry and repeated the analysis[9, 58] (Supplementary Note 9). The results showed the estimates from using the genomic regions of only NC origin (i.e., non-NC regions masked) to be largely similar to the unmasked set, suggesting that these genetic distances are inherent to the Bantu-speaking populations and not only due to differential KS admixture (Supplementary Fig. 10 and Supplementary Note 9).

**Analysis of runs of homozygosity**. The distribution of the runs of homozygosity (ROH) segments in the SAHGP populations was compared to various populations from Africa and other continents (Supplementary Note 10). The comparison of ROH between African and non-African populations challenges the previous observations of uniformly lower ROH in Africans[59, 60] and shows extreme diversity in ROH segments among African populations[30] (Fig. 4, Supplementary Fig. 11, Supplementary

Table 12, and Supplementary Data Set 1). The southern African Bantu-speakers (shown in *light purple* in Fig. 4) were found, in general, to harbor longer and more abundant ROH segments in comparison to Bantu-speaking populations from the East, Central West, and West Africa. The KS exhibited large variations in ROH length and abundance. More northern KS populations, SWB, and the Somali were found to show the highest ROH length and abundance within the continent, in some cases comparable to non-African populations. The COL, along with other recently admixed populations like the ASW and ACB, shows the lowest total ROH as well as the smallest number of segments among the African populations (Fig. 4 and Supplementary Fig. 12), reflecting their relatively recent and complex multi-ancestral admixture[13]. The significance levels of differences in ROH length between populations were estimated using the Mann–Whitney $U$-test and are shown in Supplementary Fig. 11 and Supplementary Data Set 1.

**Distribution of mitochondrial and Y chromosome haplogroups**. Mitochondrial and Y chromosome haplogroups showed a gender-biased gene flow (Table 2). The mtDNA haplogroups were predominantly KS with two-thirds showing the L0d haplogroup in both the COL and black South Africans. Two of the COL individuals had Southeast Asian/European haplogroups and one had a haplogroup found in Bantu-speakers. Four of the black South Africans had mtDNA haplogroups found among other eastern Bantu-speakers and one a haplogroup common in West Africa. Conversely, the Y haplogroups showed significant differences between the COL group (predominantly of European origin, with one African haplogroup) and black South Africans with the latter having almost exclusively African haplogroups. The self-reported black South African (XHD) with significant recent admixture had a paternal lineage of Mediterranean origin (Table 2). The mtDNA and Y haplogroup findings are consistent with previous studies that indicated cross-cultural assimilation, favoring the inclusion of female hunter–gatherers into Bantu-speaking farming communities[3, 41, 61–64].

## Discussion

The SAHGP study is the first report on the genetic architecture of Africans using high-coverage WGS data that is fully funded by an African government and analyzed and interpreted locally. It demonstrates capacity for genome analysis and highlights the high discovery rate of novel variants and a deeper understanding of population histories and affinities.

Although hinted at in an earlier study[39], population differentiation among the SEB has not been reported in any of the recent genome-scale studies. In fact, many of these studies have shown and/or assumed the SEB to be a genetically homogeneous population[30, 38]. Despite the small number of samples, our study is the first genome-scale study to report genetic differentiation between the two major language divisions of the SEB in South Africa. We postulate that one reason is the locations from which participants were sampled. In our study, we purposely recruited the SEB from rural areas or regions with little ethnolinguistic diversity, whereas other studies may have recruited from urban settings with a multi-ethnic and multi-cultural mix of individuals. Careful scrutiny of the PCA plots in the AGVP study[9], in the light of our findings, shows evidence of a tighter and more homogeneous clustering of the Zulu from Kwa-Zulu Natal and a more diffuse clustering of the Sotho who were recruited from urban Soweto, just outside Johannesburg. The latter self-reported as Sotho-speakers but may have had parents from two different ethnolinguistic groupings. Furthermore, in the detailed analysis of admixture reported in the AGVP study (Extended data Figure 7)[9] clear differences in the nature, source, and timing of admixture in the Sotho and Zulu are evident.

A failure to detect genetic differences between SEB groups who speak different but related languages in some of the previous studies is likely due to large-scale demographic changes that have occurred over the last two centuries[23]. These include migrations, displacements, admixture, and adoption of new languages, that might have rendered language alone an inadequate proxy for capturing underlying genetic differences, especially in urban centers. According to oral history, linguistic and archaeological evidence, a common ancestry is likely to be as recent as 1000–1200 years for the SOT and XHS[3, 65, 66]. Therefore, the differences in their genetic structure, in addition to differential admixture, could represent the consequences of very recent geographic, linguistic, and cultural separation with concomitant genetic drift effects, given the small effective population sizes[15, 16, 39, 67]. The small sample sizes for this pilot study, as well as the lack of population-scale WGS data from the KS populations, restricted our ability to investigate the role of genetic drift and selection in the genetic differentiation. Studies based on larger sample sizes will be necessary to assess the extent to which these factors have influenced the genomic differences. Given that differences were observed, this provides a compelling argument for investigating population substructure in South African studies as this may affect the outcomes and interpretation of biomedical genetic association and pharmacogenomics studies in the region.

Turning our attention to the admixed COL populations of South Africa, several studies have detected up to five distinct ancestry components, arising from KS, Bantu-speaker, East Asian/Southeast Asian, South Asian, and European admixture[12–14, 30, 41, 51]. In most of these studies, the Chinese and the Gujrati populations from the HapMap data set[68] have been used to represent East/Southeast Asian and South Asian ancestries, respectively. A survey of the seventeenth century slave-trade routes, however, suggests these to be unsuitable proxies for the populations that might have contributed the East Asian and South Asian ancestry in COL individuals. Based on data from the KGP[46] and Malay genome studies[50], we were able to identify the Malay as a better proxy for the Southeast Asian and the Bengali

(BEB) for the South Asian ancestry. Moreover, the geographic locations of these populations were found to be much closer to the seventeenth century Dutch trading posts[69] and historical accounts of the presence of these groups in the Cape during that time is also well documented[70]. Based on the use of the more appropriate comparative populations, we were able demonstrate that the South Asian contribution was higher in comparison to the East Asian contribution. This was corroborated in an independent study[30] of COL individuals in South Africa.

In conclusion, the SAHGP pilot study emphasizes the high discovery rate of novel variants in African populations. Despite previous reports of relatively low genetic divergence among SEBs, we detected significant population differentiation between two SEB groups in South Africa, highlighting the need to consider population structure in disease-association studies involving southern African populations. Our study is limited by the small number of participants and lack of representation of additional ethnolinguistic groups in the region. In particular, the absence of population-scale WGS data for KS groups restricted our ability to fully utilize our WGS data in analyses such as admixture mapping and local ancestry detection. The availability of such data would enable a more comprehensive analysis and is expected to provide novel insights.

## Methods

**Participants and sample collection and DNA extraction**. The study was approved by the Human Research Ethics Committee (HREC; Medical) of the University of the Witwatersrand, Johannesburg (Protocol number: M120223). Three groups of participants were enrolled and venous blood was collected into tubes containing EDTA anticoagulant. Inclusion criteria were as follows: male, over the age of 18 years, four grandparents who speak the same language as the participant (in the Bantu-speakers in order to avoid recently admixed individuals), not known to be related to the other participants in the study, and willing to provide broad informed consent (including consent to share data and DNA for future studies approved by the HREC (Medical)). Where feasible, community engagement preceded enrollment. Three main ethnolinguistic groups were included in this SAHGP pilot study. Individuals self-identified in terms of the ethnolinguistic group as part of the recruitment process. Group 1: individuals of mixed ancestry (referred to as COL in the South African context) were recruited through the Western Province Blood Transfusion Service by Sister Debbie Joubert under the guidance of Professor Soraya Bardien. Group 2: Sotho (Sotho-Tswana-speakers): seven individuals in this group were recruited from in and around the town of Ventersburg in the Free State Province, following community engagement done by Professor Michèle Ramsay and recruitment by Mr. and Mrs. Botha and Mrs. van den Berg. Group 3: Xhosa-speakers (Nguni language): eight individuals were recruited by Dr Nomlindo Makubalo from her medical clinic in the Eastern Cape Province. One individual was a Zulu-speaker (Nguni language) from Johannesburg. All DNA samples were extracted in the same laboratory using a modification of the salting out procedure[71]. The DNA was normalized and sent to the service provider (Illumina Fast Track) as a single batch at the same time, and all the data were returned in one batch.

**Data generation and processing**. The DNA samples were normalized to ~60 ng/µl and ~5 µg DNA was submitted to the Illumina Service Centre for sequencing on the Illumina HiSeq 2000 instrument (~100 bp paired-end reads, ~314 bp insert size) with a minimum of ×30 coverage. Initial analysis of the raw read data was conducted by Illumina FastTrack Sequencing Services using their in-house-developed Isaac analysis pipeline.

**SNP array data**. Each sample was also genotyped using the IlluminaOmni2.5 genotyping array.

**Whole-genome alignment and BAM processing**. Reads were aligned to NCBI 37 (hg19) of the human genome reference sequence using the Isaac Alignment Software[72]. During the mapping selection phase, low-quality 3′ ends and adaptor sequences were trimmed. Following the alignment-phase PCR duplicates were marked and indels realigned by the Isaac Alignment Software. Finally, the base-quality scores were recalibrated using GATK[73] to generate the final sorted, duplicate marked, indel-realigned BAM files that were used for variant calling (Supplementary Note 1). The quality of the alignment per sample was assessed using SAMtools version 1.1-26-g29b0367[74] to examine the percentage of duplicates and successfully mapped reads (Supplementary Table 1).

**SNV calling**. SNV calling was performed on all samples using the Isaac Variant Caller. The final data set of variants produced by the Isaac Variant Caller was filtered based on various features to generate a high-quality SNV data set (Supplementary Note 1). To assess the accuracy of the variant calls generated by the Isaac Variant Caller, two additional approaches were used to recall variants using the BAM files produced by the Isaac Alignment software. Variant calling was conducted using GATK's HaplotypeCaller version 3.2-2[74]. The variant calling was conducted independently at the University of the Witwatersrand (Wits) and the University of Pretoria (UP) using the same GATK pipeline with varying parameters (Supplementary Note 1). The Wits site conducted the variant calling using GATK's suggested best practices, while UP used more stringent variant-calling parameters (Supplementary Note 1). Each of the GATK variant call data sets was filtered using the GATK Variant Quality Score Recalibration and the transition–transversion ratios assessed across the range of MACs (Supplementary Note 1, Supplementary Table 2, and Supplementary Fig. 1). The concordance between the three filtered data sets was examined and found to have an overlap of 97% for the SNVs called (Supplementary Fig. 1). In order to move forward with a high quality, robust set of SNVs, the intersection of filtered SNVs called by all three approaches was used for further downstream analysis.

**Indels and structural variant calling**. Indels and structural variants were called using the Isaac variant caller software according to the Illumina pipeline[72].

**Functional categories for SNVs and indels**. The annotation was performed with the ANNOVAR software[45] using the database version (2015Mar22). Variant type counts for SNVs, indels and CNVs within each population was calculated.

**Gene descriptions**. The identification of genes in genomic regions of interests was performed using BioMart (http://www.ensembl.org/biomart/). The description of genes and their potential functions was inferred using GeneCards (http://www.genecards.org/).

**Relatedness**. As several of the analysis methods used in this study assume the use of unrelated samples for accurate results, we assessed the data set for relatedness using an identity-by-descent (IBD) approach in PLINK v1.9[75]. The IBD approach is based on calculating genome-wide identity by state (IBS) for each pair of individuals, based on the average proportion of alleles shared in common at the genotyped SNPs. The genotype data set was used for the IBD analysis and revealed no level of relatedness based on the π_hat values generated, where values of greater than 0.1875 are indicative of closely related individuals.

**Site frequency spectrum**. To avoid bias due to possible incorrect assignment of ancestral alleles, a folded site frequency spectrum (SFS) based on MACs was calculated using a custom perl script. The script was used to study SFS in the three SAHGP populations along with eight randomly selected samples from representative African (YRI, ASW) and non-African populations (CEU) from the KGP[46] and the AGVP[9] (ZUL) data sets. As the main application of this analysis was to compare the SFS within each data set, it needs to be noted that variation in sequencing depths among data sets might introduce some biases in cross data set comparisons.

**Mitochondrial DNA haplotype calling**. Haplogrep2[76] was used to identify mitochondrial haplotypes for each individual. For this, all reads were aligned using BWA-mem to the RSRS sequence. The BAM files produced were then uploaded to mtDNA-server service as suggested by the webserver documentation. This service performs QC filtering (Mapping Quality Score < 20; read alignment quality < 30; base quality < 20; heteroplasmy level < 1%; and BAQ filtering) and annotates regions of low complexity and NUMTS and finally assigns the most likely haplogroups.

**Y chromosome haplogroup analysis**. Y-chromosome haplogroup analysis was done using the AMY-tree algorithm and tool[77]. For each person, the variants detected from the WGS were extracted, and converted into the correct format before being input into the AMY-tree program.

**LOF analysis**. The LOF mutations in our data set include Stop Gain, Stop Loss, Frameshifts (defined as indel in exon which in not a multiple of 3), and Splice Variants (defined as SNP/indel in position +1, +2, −1, −2 in introns). The above-mentioned categories of mutations in the whole-genome sequence data were identified using ANNOVAR[45]. The SNVs showing MAF > 0.05 were excluded as they were assumed to be mutations of lower impact. The distribution of the LOF variants in each individual was analyzed and if an individual was found to contain two different heterozygous LOF mutations, one in each chromosome, as inferred from phased whole-genome sequence data, in the same gene, the individual was characterized as a potential "complete knockout" with respect to that gene. Not all SNVs could be phased accurately because they were novel and therefore, when in doubt, we made the assumption that they were in a *trans*-configuration.

**Population structure and admixture and relationship analysis**. We investigated population structure using both PCA and structure analysis. In choosing comparative populations we used prior work and historical knowledge. After some preliminary experimentation we chose specific data sets (Supplementary Table 10). The KGP[46] data for Yoruba in Ibadan (YRI), Luhya in Webuye, Kenya (LWK), Utah residents with Northern and Western European ancestry (CEU), and Bengali in Bangladesh (BEB) were used. In addition, Malays from the Singapore Sequencing Malay Project[50]; Black South Africans from Soweto (SEB2)[34]; several populations from the study by Schlebusch et al.[30], namely several Khoe-San (KS), and COL groups (COLC and COLW), South-east Bantu-speakers (SEB1) and SWB were included in the comparisons. Moreover, Baganda (BAG) and Zulu (ZUL) whole-genome sequences from the AGVP data set[9] were also included in the analyses. The data were merged using PLINK v1.9[75], and filtered to exclude SNVs and/or individuals with poor quality. For both PCA and ADMIXTURE the SNVs were pruned to select sample SNVs not in LD with each other, leaving ~197 K SNVs for analysis.

**PCA plots**. PCA analysis was done using PLINK v1.9[75]. Further analysis was done using EIGENSTRAT[78] in order to estimate the statistical difference between the XHS and SOT.

**Population structure analysis**. Structure analysis was done using ADMIXTURE[79]. For $K = 3,..,10$, 40 independent runs were performed using ADMIXTURE, which were averaged using CLUMPP[80]. The minimum cross-validation score computed by Admixture is for $K = 7$. The tool Genesis (http://www.bioinf.wits.ac.za/software/genesis) was used to visualize the results from the PCA and population structure analyses.

**Population differentiation**. The analysis of the fixation index ($F_{ST}$) at the whole-genome level provides an estimate of the genetic distance between two populations and has been used extensively in inferring relationships between a set of populations[9, 81]. We investigated the relationship between the southern African populations in our data sets and two distinct sets of populations known to be related to them; the Bantu-speaking groups (from South, West, and East Africa) and the KS populations from southern Africa. For this a merged data set consisting of the SAHGP data, Schlebusch et al.[30] and AGVP[9] was generated. The Weir and Cockerham's (WC) $F_{ST}$ estimate[82] was computed between the SAHGP and other groups using PLINK v1.9[75].

**Local ancestry-based masking**. Three data sets—the SAHGP, KGP[46], and Schlebusch et al.[30], all genotyped on the Illumina Omni 2.5 M SNP chip—were merged together using PLINK v1.9[75]. The merged data set was phased using SHAPEIT2[83] with standard parameters. Analysis of local ancestry was performed using PCAdmix[58], with Ju/'hoansi, YRI, and non-African (CEU, CHB, and JPT) as the three ancestral populations and the SEB2 from Schlebusch et al.[30] as the target population. Based on the ascertainment of ancestry of all the 20 SNP windows, the windows showing <20% of YRI ancestry were masked out to generate a minimal non-admixed SEB data set.

**f2 and rare variant sharing analysis**. To compare rare allele sharing between the SAHGP and the KGP[46] data set, we merged the 15 SEB individuals (7 SOT, 7 XHS, and the ZUS) with the KGP[46] data sets and identified those variants that occur precisely twice in the merged data set ($f_2$ variants)[46, 84]. As the sample sizes in the two data sets were not uniform and an unbiased estimate of $f_2$ sharing was difficult, instead of performing a complete $f_2$ analysis we focused on those $f_2$ variants that occur at least once in one of the 15 SEB and once in the KGP[46] data set. A similar analysis was performed using SNVs shared between only two populations irrespective of the minor allele frequency. This was mainly done to compensate for the small sample size in our study, which might have considered some SNVs to be singletons that could have been present multiple times if we had included more samples. Both these analyses were performed for the COL individuals. We also identified SNPs that occur in only one of the five continental population sets in the KGP[46] data and studied their distribution in the 15 SEB and the 8 COL individuals.

**SNV density comparisons**. To study the variation in SNV enrichment patterns within Africa, we compared SNV densities in the YRI and LWK from the KGP[46] data set to Zulu from the AGVP[9] and SEB from the SAHGP data set. For this, we scanned the genome using 1 Mb sliding windows (with no overlap) and computed the number of SNVs occurring in that region in each population. The empirical distribution of SNV densities thus obtained for each population was used to assign a rank score and *p*-value to the density level observed for each window in that population. A similar scan was conducted using 25 kb windows. We noted that there are marked differences in sample size and coverage between data sets such as KGP[46], AGVP[9], and SAHGP, and these factors could also result in differences in estimation of SNV densities. Therefore, we considered only the regions for which both Zulu and SEB were found to show similar SNVs densities and vary strongly with both of the other African populations.

**ROH analysis**. Three data sets—the SAHGP, AGVP[9], and Schlebusch et al.[30], all genotyped on Illumina Omni 2.5 M SNP chip, were merged using PLINK v1.9[75] (Supplementary Table 12). An overall QC was performed on the merged data and SNVs with missingness greater than 0.05 and individuals with missingness greater than 0.05 were removed. We also excluded SNVs showing extreme deviation from Hardy–Weinberg equilibrium ($p$-value $<1 \times 10^{-7}$) from the data. The populations were merged according to linguistic and geographic affinities into superpopulations (Supplementary Table 12). To correct for possible ascertainment bias, SNVs with frequencies lower than 0.01 in any of the merged superpopulations were removed (Supplementary Table 12). This resulted in a data set containing around 500 K SNPs (total genotyping rate in this data set was 0.999182). Total ROH length and number of ROH segments were estimated using PLINK v1.9[75]. By default, in PLINK v1.9 only ROH containing at least 100 SNVs, and of total length ≥1000 kb are noted. Therefore, we performed an additional analysis with the ROH window size set to 500 kb. The scanning window contained 50 SNVs and a scanning window hit was allowed to contain at most one heterozygous call and five missing calls. The Mann–Whitney $U$-test was used to test differences between the total lengths of ROH distribution in population and superpopulation pairs.

**Regions of extreme differentiation between Sotho and Xhosa**. To identify regions that show high $F_{ST}$ variation within the SOT and XHS, the WC $F_{ST}$ estimate for each SNV was computed using PLINK v1.9[75]. A sliding window of 25 kb was used to scan the distribution of average $F_{ST}$ scores across the genome and the top 0.005% windows showing highest $F_{ST}$ scores were identified. As the WGS data include a lot of novel and population-specific SNVs, only the windows containing at least 10 SNVs that were found to be present in both the populations were considered (Supplementary Table 11).

**Novel SNV identification and their genomic distribution**. The novel SNVs reported in this study were identified by comparing the presence of the SNVs occurring in the 15 SEB samples to all SNPs in the dbSNP142[48], in the KGP[46] and AGVP[9]. To identify genomic regions enriched in novel SNVs, a sliding window of 1 Mb was used to scan the genomes and the regions showing most number of novel SNVs were selected. A similar data set was generated using 25 kb sliding windows.

**Data availability**. The WGS and the SNP-array data that form the basis of the findings reported in the study have been deposited in the in the European Genome-phenome Archive (EGA; https://www.ebi.ac.uk/ega/home; accession numbers: study: EGAS00001002639, sequence data set: EGAD00001003791, array data set: EGAD00010001418). Access to data is determined by a Data Access Committee (DAC: EGAC00001000734). Data access decisions can be passed to the EGA by emailing ega-helpdesk@ebi.ac.uk with the email address of each applicant and confirmation of the dataset(s) to provide access. The EGA will then create an EGA account with the relevant access permissions.

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

## Acknowledgements

We are grateful to the South African National Department of Science and Technology for funding this initiative under the umbrella of the Southern African Human Genome Programme (SAHGP). We thank Sister Debbie Joubert and the Western Province Blood Transfusion Service for recruitment of the COL study participants, Dr Nomlindo Makubalo for recruitment of the Xhosa-speakers from the Eastern Cape and Mr. and Mrs. Botha and Mrs. van der Berg for recruitment of the Sotho-speakers from the Free state Province, as well as all the participants for generously agreeing to share their data and biological samples. We also thank the participants of the SAHGP launch meeting held in 2011 from which this pilot project was initiated (Supplementary Note 11). A.C. was supported by the AWI-Gen Collaborative Centre funded by the NIH (1U54HG006938) as part of the H3Africa Consortium. M.R. is a South African Research Chair in Genomics and Bioinformatics of African populations hosted by the University of the Witwatersrand, funded by the Department of Science and Technology and administered by National Research Foundation of South Africa (N.R.F.). M.S.P. was funded by the South African Medical Research Council (Flagship and Stem Cell Extra-mural Unit awards) and the Institute for Cellular and Molecular Medicine (University of Pretoria). N.M. and S.A. were supported by the H3ABioNet NIH grant (U41HG006941).

## Author contributions

M.R. and M.S.P. co-lead the SAHGP initiative, and the project was designed and coordinated by the core working group including M.R., M.S.P., S.B., H.S., R.R., J.R., K.S., P.V., N.M., F.J., S.H., and L.V. M.R. and H.S. obtained ethics approval for the study. The data analysis team was led by S.H. (PCA; STRUCTURE and Y chromosome analysis) and included A.C. (novel SNV characterization, LOF variant, $f_2$, $F_{ST}$, SFS, and ROH analysis), N.M. (functional analysis), F.J. (variant calling), S.A. (variant calling), G.B. (functional annotation and data curation), E.R.C. (admixture), J.G. (functional annotation), M.J.S.D. (functional annotation), A.M. (functional annotation, SNV characterization, data cura-tion, and mtDNA analysis), and D.S. (regional $F_{ST}$ analysis, data visualization). All authors wrote the Methods section and notes on their analyses. M.R. and A.C. drafted the manuscript, and A.C. was responsible for coordinating Tables and Figures (including the Supplement). All authors read, commented on, and approved the manuscript.

## Additional information

**Competing interests:** The authors declare no competing financial interests.

