## [Peer Review File · Nature Communications]

Reviewers' comments:

Reviewer #1 (Remarks to the Author):

GENERAL COMMENTS:

The paper is a pilot study to characterise 24 genomes from South Africa, in particular 16 black SouthEastern Bantu-speakers. They seek to understand these samples' contribution to human genetic diversity. The samples selected are of great interest and it would be exciting to see the work develop.

However, the paper as it stand seems weak, and lacking in detail in many crucial places. The processing procedure followed is poorly described. It seems likely that the variants identified are false positives and there is no evidence to the contrary, and there is no discussion of filtering of variants, supported by QC metrics on the variants found. In addition, there is no indication that the alignment files and vcf files have been submitted to a public repository such as the European Nucleotide Archive, EVA or dbgap. Therefore the data appears unobtainable, the processing performed difficult to replicate, and observations and conclusions hard to reproduce.

SPECIFIC COMMENTS:

The authors:

- find 16 M unique variants, 0.8M novel SNVs,
- find that despite the shallow time-depth between the two Southeastern Bantu speaking groups (Nguni and Sotho-Tswana), there is: (i) strong differentiation, (ii) regions of high divergence.

Aside: presumably these regions of high divergence are driving the separation (which could be checked from snp loadings, but seems likely). These regions could be characterised.

(A) Concerns on data production and availability

(A1) Data availability: Download details for the bams and vcfs (variant files) should be provided.

(A2) Consistency of analysis:

Was a consistent wetlab and informatics procedure used for all samples? Is there any possibility of batch effects that might explain the differences observed?

(A3) Alignment details are extremely terse: The authors write: "Variants were called according to the GATK best practices for variant calling from cohorts of samples, using the BAM files generated by the Isaac Aligner.". This is not reproducible, because the 'best practices' change with new developments. How many raw reads were generated for each sample? Were duplicates removed (if so how many) - what procedure was followed? What software? What versions of the software? Were quality scores recalibrated? Was indel-realignment performed, which should lead to more reliable genotypes and indels? Ideally a table summarising each of these steps, in counts of reads for each sample, should be provided.

It would also be useful to know if adapter trimming was performed. If it were not, this could lead to false genotypes to be called.

(A4) Variant filtering:

There is no discussion of filtering of variants. Surprisingly, the word 'filter' does not appear in the supplementary 1 where one might expect to see this, and in fact appears once and once only in all the supplementaries. This is typically a huge component of variant analysis, and can be very challenging. Filtering is mentioned in Supplementary 4 for PCA analysis which does not affect the

raw counts of variants found. Alignments have been generated using a particular aligner (Isaac), and then variants are called using essentially two independent methods (the method used by University of Pretoria is likely to only generate a subset of the variants from that of the University of Witwatersrand). Taking an intersection of the 3 (well, two really) methods is not adequate to remove false positives. Common filters to use would be mappability filters, tandem repeat filters etc, which help eliminate artifact variants which arise from common misalignment regions. False positives could easily pass through both genotyping methods and would inflate the 'number of unique variants found from these samples'.

Therefore, I would suggest minimally:

(A4.1) a venn diagram of the 3 variant calling methods, showing counts for method specific methods,

(A4.2) Transition-transversion metrics. The vast majority of new variants according to Figure 1(f) have MAC=1 which is understandable given the low sample count. However, there is no indication of whether these are real. Transition-transversion metrics of the variants, both for the entire set and stratified by MAC would indicate if these are true variants. Ideally, Ts-Tv rates as a function of filters used would be good. I note that the genotype concordance with the Omni chip is reported at 99.5% however, this tells us nothing about the variants called at sites which are not on the genotyping array. And a PCA analysis does not help, because it is not sensitive to random noise from false positives.

(A4.3) specify the specific parameters used for variant calling in the supplementary text (the authors have noted that these are available on request), but it is better to report them as they are not typically verbose.

Another possibility might be to look at the within-population allele-frequency spectrum. An additional possibility is to do a PSMC analysis of variants; an inflated recent population size would point to inadequate filtering.

(A4.4) Question: Variants for 1kg with MAF>0.01 were removed, which means that lower frequency variants found in 1kg are ignored. This might also inflate the number of new variants reported. Could the authors comment on the choice of this threshold and what would happen if it were lowered?

(B) Concerns about analysis

Given the lack of details about variants, analytical conclusions are worrisome, and might be artifacts. Some specific concerns follow.

(B1) Criticism of MT handling. The authors note that: "For each sample, reads were first aligned to the GATK reference bundle. Reads mapping to MT were then re-aligned to the Reconstructed Sapiens Reference Sequence (RSRS)⁴³ and genotypes called using GATK Haplotypecaller. The calls were compared to RSRS to create a list of mutations for the sample. These mutations were then compared to each terminal node in Phylotree version 1444 and the terminal node with the most matching parents was chosen as the most likely haplogroup." (p7 Main paper).

This procedure is problematic in several ways:

i) numts (mitochondrial sequences homologous to autosomal sequences) are ignored, which leads to read loss and potential bias in the mt coverage. There are 755 known numts, ranging in size from 39bp to almost the whole mt sequence. A better procedure is to either align all reads to the RSRS sequence (which is fairly quick because the MT genome is small), or less preferable, to identify all reads from both the numts and MT genome, and align these to the RSRS.

ii) MT haplogroup calling here seems to have been done in a manual (and potentially error-prone) way, as I don't believe phylotree provides haplogroup assignment. If I understand the authors correctly - it would be better to use an automated tool such as: Haplogrep, mthap, or Haplofind.

(B2) f2 variant analysis is a fairly recent technique and should be referenced, eg: Mathieson I, McVean G (2014) Demography and the Age of Rare Variants. PLoS Genet 10(8): e1004528. doi:10.1371/journal.pgen.1004528 or similar.

Could the authors comment on Figure 3 of the main paper: in the f2 analysis, we should expect that the COL population which is a known mixture of Indian and other populations, should share more f2 variants with 1kg south asian populations such as GIH and BEB, yet it seems that there are actually fewer compared with the SOT sample. Does this seem surprising?

(B3) Details missing in Fst analysis:

n

In the Fst analysis (Section 9, supplementaries), an important analysis to measure the difference between populations, it would be useful to see details. For examples, the authors write: "For this, a merged dataset constituting of data from the SAHGP, AGVP and Sclebusch et al. 2012 studies^{4,7} was generated.". However, there's no mention of the number of snps that result, the number that are removed and filtering (based on say, missingness) is not mentioned.

(B4) Regions of divergence between the Bantu speaking groups.

This was interesting, and section 6 was good

(B5) Supplementary Table1: Sequencing Coverage observed in the 24 samples.

The XHS samples have consistently lower coverage than almost all the other samples - do the authors believe this is due to reference bias, given the hg19 reference is a mixture of other varying groups which do not include XHS.

Minor typo:

"Supplementary Table 13: P-values^r for differences in total ROH lengths of individuals from all 49 populations" (in the word 'P-values')

In conclusion, it would seem a substantial rewriting needs to be done.

Reviewer #2 (Remarks to the Author):

Strengths of this manuscript are variant discovery in understudied ancestral populations and local genomic capacity development in a low resource environment. There are however some weaknesses the most significant being the small number of individuals sequenced and the methodologies used to analyze generated sequence data. The analysis basically reduced the sequence data to genotype by not implementing the latest analytic strategies for the analysis of sequence data – for example, the Pairwise Sequentially Markovian Coalescent model or the Multiple Sequentially Markovian Coalescent model for the analysis of whole genome sequences to infer human population history. In addition, it is not immediately apparent what novel population

genetics insights were provided by the sequencing of these small number of individuals that we do not already know.

In order to provide context for whole genome sequencing, the Introduction should refer to SC Schuster et al. (2010 Nature 463:943-947) and HL Kim et al. (2014 Nat Commun 5:5692). The published literature contains papers not cited in this manuscript that address ancestry in relevant groups of South Africans, e.g., E de Wit et al. (2010 Hum Genet 128:145-153), D Shriner et al. (2014 Sci Rep 4:6055), and ER Chimusa et al. (2015 PLOS Genet 11:e1005052).

Looking at Figure 2C, it appears that the SOT and XHS are essentially two-way mixtures of ancestries shared among Khoisan (shown in light blue) and Southeastern Bantu (shown in red) speakers, although there are no error bars. Thus, it appears that the ancestral difference between the SOT and XHS is not qualitative but quantitative. It is not clear whether differentiation between the Nguni and Sotho-Tswana speakers reflects (1) different mixture proportions, with the SOT having a slightly higher percentage of Khoisan ancestry, or (2) divergence due to random genetic drift from a common ancestor. Similarly, in Figure 2D, it appears as though the trend in F_{st} is simply recapitulating the trend in mixing proportions, with the Khwe having the least Khoisan ancestry and the Ju/'hoansi having the most Khoisan ancestry. Note that differences in mixing proportions violates the assumptions of Weir and Cockerham's F_{st} .

The authors correctly point out the problem in the f_2 analysis given the unequal sample sizes between the 1000 Genomes data and their own data. I recommend randomly drawing subsets of individuals from the 1000 Genome data.

P. 2: The statement regarding the number of singletons is puzzling. Singletons are always expected, regardless of sample size. Do the authors mean that the number of singletons was not different from the expected value, given the sample size?

In Table 2, what data suggest that any Y haplogroup E has a K-S origin?

In Figure 1d, indels is misspelled in the y-axis label.

Responses to reviewer comments:

Reviewer #1 (Remarks to the Author):

(A) Concerns on data production and availability

(A1) Data availability: Download details for the bams and vcfs (variant files) should be provided.

Response: WGS and Genotyping Array Data will be available through application on the SAHGP website (URL: <http://saghp.sanbi.ac.za/index.php>). We would be happy to provide a link to our server at the University of the Witwatersrand for the reviewers to access that data for review purposes. We are awaiting ethics approval to activate the link to the data request process. The data can also be “searched” through the Global Alliance for Genomics and Health “Beacon project”.

(A2) Consistency of analysis: Was a consistent wetlab and informatics procedure used for all samples? Is there any possibility of batch effects that might explain the differences observed?

Response: The DNA was extracted in the same laboratory using a modification of the salting out procedure. The DNA was normalized and sent to the service provider (Illumina Fast Track) as a single batch at the same time and all the data were returned in one batch.

(A3) Alignment details are extremely terse: The authors write: "Variants were called according to the GATK best practices for variant calling from cohorts of samples, using the BAM files generated by the Isaac Aligner.". This is not reproducible, because the 'best practices' change with new developments. How many raw reads were generated for each sample? Were duplicates removed (if so how many) - what procedure was followed? What software? What versions of the software?

Response: A table specifying the number of reads and duplicates marked has been added as Supplementary Table 1a. The procedure was added to the online methods and details are available in the supplementary note.

Were quality scores recalibrated? Was indel-realignment performed, which should lead to more reliable genotypes and indels? Ideally a table summarising each of these steps, in counts of reads for each sample, should be provided. It would also be useful to know if adaptor trimming was performed. If it were not, this could lead to false genotypes to be called.

Response: Information about the base quality score recalibration and indel realignment and adaptor trimming was added to online methods and supplementary note. The parameters for the scripts for the variant calling have been added to the supplementary note.

(A4) Variant filtering: There is no discussion of filtering of variants. Surprisingly, the word 'filter' does not appear in the supplementary 1 where one might expect to see this, and in fact appears once and once only in all the supplementaries. This is typically a huge component of variant analysis, and can be very challenging. Filtering is mentioned in Supplementary 4 for PCA analysis which does not affect the raw counts of variants found. Alignments have been generated using a particular aligner (Isaac), and then variants are called using essentially two independent methods (the method used by University of

Pretoria is likely to only generate a subset of the variants from that of the University of Witwatersrand). Taking an intersection of the 3 (well, two really) methods is not adequate to remove false positives. Common filters to use would be mappability filters, tandem repeat filters etc, which help eliminate artifact variants which arise from common misalignment regions. False positives could easily pass through both genotyping methods and would inflate the 'number of unique variants found from these samples'.

Response: Details on filtering of variants has been added to the Online Methods and the Supplementary Notes.

Therefore, I would suggest minimally:

(A4.1) a venn diagram of the 3 variant calling methods, showing counts for method specific methods,

Response: The Venn diagrams with counts per sample have been included in Supplementary Figure 1c.

(A4.2) Transition-transversion metrics. The vast majority of new variants according to Figure 1(f) have MAC=1 which is understandable given the low sample count. However, there is no indication of whether these are real. Transition-transversion metrics of the variants, both for the entire set and stratified by MAC would indicate if these are true variants. Ideally, Ts-Tv rates as a function of filters used would be good. I note that the genotype concordance with the Omni chip is reported at 99.5% however, this tells us nothing about the variants called at sites which are not on the genotyping array. And a PCA analysis does not help, because it is not sensitive to random noise from false positives.

Response: A plot of the TsTv ratio as a function of minor allele count has been added as Supplementary Figure 1b. All ratios were in the acceptable range of 2 - 2.2. A comment was added to online methods and supplementary text. A table of TsTv ratio per sample stratified by variants in dbSNP_183 and novel SNPs has been added as Supplementary Table 2b.

(A4.3) specify the specific parameters used for variant calling in the supplementary text (the authors have noted that these are available on request), but it is better to report them as they are not typically verbose. Another possibility might be to look at the within-population allele-frequency spectrum. An additional possibility is to do a PSMC analysis of variants; an inflated recent population size would point to inadequate filtering.

Response: Details were added in the supplementary text regarding - Raw read processing, BQSR, reason for not doing indel realignment, HC parameters, VQSR parameters. We would be happy to add a link to scripts that we can place online if necessary. We have re-written the relevant sections in the Online methods and Supplementary Notes.

We have added an allele frequency spectrum analysis as suggested that shows the number of sites in each MAF bin largely corresponds to the expected values based on random African sample WGS sample sets of same size (and added Figure 1f) and described the observations in the Supplementary Note.

(A4.4) Question: Variants for 1kg with MAF>0.01 were removed, which means that lower frequency variants found in 1kg are ignored. This might also inflate the number of new

variants reported. Could the authors comment on the choice of this threshold and what would happen if it were lowered?

Response: Thanks for highlighting the confusion. These variants were not removed from the core data at any point but were removed for specific analyses such as PCA and admixture. This filtering was therefore only performed for the PCA and ADMIXTURE analysis, to reduce the impact of low-frequency and dataset specific variants. We completely agree with the reviewer's concern and would like to confirm that we did not use this filtering for any of the other analyses. This is now clearly documented in the text.

(B) Concerns about analysis

Given the lack of details about variants, analytical conclusions are worrisome, and might be artifacts. Some specific concerns follow.

(B1) Criticism of MT handling. The authors note that: "For each sample, reads were first aligned to the GATK reference bundle. Reads mapping to MT were then re-aligned to the Reconstructed Sapiens Reference Sequence (RSRS)⁴³ and genotypes called using GATK Haplotypecaller. The calls were compared to RSRS to create a list of mutations for the sample. These mutations were then compared to each terminal node in Phylotree version 1444 and the terminal node with the most matching parents was chosen as the most likely haplogroup." (p7 Main paper). This procedure is problematic in several ways: i) numts (mitochondrial sequences homologous to autosomal sequences) are ignored, which leads to read loss and potential bias in the mt coverage. There are 755 known numts, ranging in size from 39bp to almost the whole mt sequence. A better procedure is to either align all reads to the RSRS sequence (which is fairly quick because the MT genome is small), or less preferable, to identify all reads from both the numts and MT genome, and align these to the RSRS. ii) MT haplogroup calling here seems to have been done in a manual (and potentially error-prone) way, as I don't believe phylotree provides haplogroup assignment. If I understand the authors correctly - it would be better to use an automated tool such as: Haplogrep, mthap, or Haplofind.

Response: As suggested by the reviewer we have used Haplogrep2 to repeat the mtDNA haplogroup analysis and essentially identified the same haplogroups, with one exception. This, however, did not affect the nature of the origins of the mtDNA lineages and the main conclusion stays the same. The results, methods and references were updated accordingly.

(B2) f2 variant analysis is a fairly recent technique and should be referenced, eg: Mathieson I, McVean G (2014) Demography and the Age of Rare Variants. PLoS Genet 10(8): e1004528. doi:10.1371/journal.pgen.1004528 or similar. Could the authors comment on Figure 3 of the main paper: in the f2 analysis, we should expect that the COL population which is a known mixture of Indian and other populations, should share more f2 variants with 1kg south asian populations such as GIH and BEB, yet it seems that there are actually fewer compared with the SOT sample. Does this seem surprising?

Response: As suggested by the reviewer we also expected to see some of the other ancestries in the COL to be captured in the f2 analysis. Therefore, while describing this in the supplementary notes, we have included a few lines stating this and speculated on the possible reasons for such an observation (Supplementary Notes Page 9 last paragraph). We also performed an overall SNP sharing analysis to minimize the loss of real f2 variants due to

sample size differences. The results are summarized in Supplementary Figure 9, and were similar to that observed for f_2 analysis.

We believe that the problem is that f_2 analysis has not been designed to detect admixture as seen in the COL. A scenario where a variant is shared between a population and a distant offshore relative (due an historical admixture event) but not with any neighboring present day population from the same geographic region, might be problematic. For example, in the Phase 3 KGP analysis we do not see any significant second or subsequent component for known admixed populations such as ACB ASW, MXL, and BEB. Therefore, the major ancestry in the Southern African groups, as expected, was the East African Bantu which makes sense in terms of known models of Bantu migration as well as history.

An intuitive alternative to this appeared to be the study of SNPs that are shared by populations in a continent and its offshore relatives. We have provided an additional analysis aimed at studying continent-specific SNP sharing rather than population-specific SNP sharing (as implemented in f_2). The results from this analysis, integrated into Supplementary Figure 9, clearly shows that analysis of continent-specific SNP sharing yields similar results. A line has been added to the main text and a paragraph has been added to the supplementary note to describe this analysis.

(B3) Details missing in F_{ST} analysis: In the F_{ST} analysis (Section 9, supplementaries), an important analysis to measure the difference between populations, it would be useful to see details. For examples, the authors write: "For this, a merged dataset constituting of data from the SAHGP, AGVP and Sclebusch et al. 2012 studies 4,7 was generated. ". However, there's no mention of the number of snps that result, the number that are removed and filtering (based on say, missingness) is not mentioned.

Response: We thank the reviewer for pointing out this gap in the methods section. A paragraph has been added to the online methods section and additional references were added (Page 9 first paragraph) (see below).

"For merging the above-mentioned datasets genotyped on Omni 2.5M SNP chip, SNPs that has been successfully genotyped in all the three studies were identified. The genotype data corresponding to these SNPs were extracted from each dataset and merged using PLINK1.9. The dataset thus generated contained 1,028,376 SNPs and 3,108 individuals. After removing SNPs that show discordance in alleles, filtering based on both individual level missingness of >0.05 , SNP level missingness of >0.05 , and HWE cut of $p < 0.0000001$, retained 1,026,664 SNPs and 3,108 individuals for F_{ST} analysis."

(B4) Regions of divergence between the Bantu speaking groups. This was interesting, and section 6 was good

Response: Thanks.

(B5) Supplementary Table1: Sequencing Coverage observed in the 24 samples. The XHS samples have consistently lower coverage than almost all the other samples - do the authors believe this is due to reference bias, given the hg19 reference is a mixture of other varying groups which do not include XHS.

Response: We appreciate the reviewer's careful scrutiny of the table. Statistical tests

support this observation, to some extent. The t-test based P-value for the differences between mean coverage in COL and XHS is <0.001 . However, the P-value for differences in COL and SOT is not significant (0.8). Moreover, the P-value of difference between SOT and XHO is also not significant, even if we remove the SOT individual with very low coverage. Therefore, while there is a slight trend in the data and reference bias seems a logical source for the observed differences in coverage, the evidence in the data is not strong enough to form the basis of any hypothesis. We have added a paragraph in the supplementary note stating this (as follows).

“Although we observed that the overall coverage in the XHO samples was lower than in the other two populations a t-test based evaluation showed the differences to be statistically significant only in the XHS-COL, but not in the other two comparisons (SOT-COL and XHS-COL). Therefore, the observed differences do not appear to be related to demographic history or ancestry.”

Minor typo: "Supplementary Table 13: P-valuesr for differences in total ROH lengths of individuals from all 49 populations" (in the word 'P-values').

Response: Corrected

Reviewer #2 (Remarks to the Author):

Strengths of this manuscript are variant discovery in understudied ancestral populations and local genomic capacity development in a low resource environment. There are however some weaknesses the most significant being the small number of individuals sequenced and the methodologies used to analyze generated sequence data.

Response: The methods are now described in much more detail and in line with the recommendations from reviewers 1 and 2.

The analysis basically reduced the sequence data to genotype by not implementing the latest analytic strategies for the analysis of sequence data – for example, the Pairwise Sequentially Markovian Coalescent model or the Multiple Sequentially Markovian Coalescent model for the analysis of whole genome sequences to infer human population history.

Response: We acknowledge this limitation. The major factor that made us reduce the data for some of the analyses (PCA and STRUCTURE) to chip data, as the reviewer has pointed out, is the unavailability of population level WGS from southern African hunter-gatherers (the whole genome data in Lachance et.al. 2012 were from eastern and central Africa. The southern African hunter-gatherer data in Kim et al. 2014 included only 2 individuals per group from two hunter-gatherer groups and therefore were insufficient for most analyses). Population level datasets from southern African hunter-gathers will enable future researchers to perform many of these studies at the WGS level.

We also completely agree about the MSMC analysis. There were a couple of issues, which did not allow us to include the analysis in the first draft. Firstly, although being widely used, the tool is quite complex in terms of integration of data from independent studies and also computationally intensive. Moreover, the high Khoesan admixture in the SAHG populations and its effect on MSMC based estimates needed evaluation. We have set up

collaboration with a research group with expertise in this area and initiated the analysis. We hope to be able to include this analysis if given an opportunity to resubmit.

In addition, it is not immediately apparent what novel population genetics insights were provided by the sequencing of these small number of individuals that we do not already know.

Response: The novelty is that we found significant differentiation between the SOT and XHO who have only been geographically separated for ~1200 years. We also provide analyses and discussion on the potential drivers for this divergence. In addition, we have shown that despite the fact that there is now a lot more WGS data available, we still detected ~0.8M novel SNVs. We have shown that these variants are not equally distributed throughout the genome. Moreover, this will be the first **high-coverage WGS data** from Southern African Bantu-speakers and also the highly admixed coloured population and would provide future researchers an important dataset for more detailed analysis of admixture and local ancestry as the hunter-gatherer WGS data from this region becomes available.

In order to provide context for whole genome sequencing, the Introduction should refer to SC Schuster et al. (2010 Nature 463:943-947) and HL Kim et al. (2014 Nat Commun 5:5692). The published literature contains papers not cited in this manuscript that address ancestry in relevant groups of South Africans, e.g., E de Wit et al. (2010 Hum Genet 128:145-153), D Shriener et al. (2014 Sci Rep 4:6055), and ER Chimusa et al. (2015 PLOS Genet 11:e1005052).

Response: Reference to these studies has been included and the studies have been given context.

Looking at Figure 2C, it appears that the SOT and XHS are essentially two-way mixtures of ancestries shared among Khoisan (shown in light blue) and Southeastern Bantu (shown in red) speakers, although there are no error bars. Thus, it appears that the ancestral difference between the SOT and XHS is not qualitative but quantitative. It is not clear whether differentiation between the Nguni and Sotho-Tswana speakers reflects (1) different mixture proportions, with the SOT having a slightly higher percentage of Khoisan ancestry, or (2) divergence due to random genetic drift from a common ancestor.

Response: As suggested by the reviewer, the differences between the SOT and XHS seem largely quantitative and suggest possibility (1) as the source of the difference between the two populations. However, we agree that it is also important to investigate whether the alternative (2) also plays some role in the observed differentiation of these populations. Site frequency spectrum comparisons have been shown to correspond to, and vary according to, genetic drift at least between various continental populations (Keinan et al. 2007). The high similarity in the site frequency spectrum of SOT and XHS suggests that there has been little drift after separation in these populations. Similarly, we could not find any notable difference in the level of heterozygosity, heterozygosity/homozygosity ratios, IBD or inbreeding coefficient, which has also been suggested to accompany genetic drift. To explore the role of genetic drift further, we also performed an out-group f_3 analysis, which again showed little evidence of genetic drift.

As we were not able to find a method that has been shown to be able to discern the difference between drift and admixture in populations diverged around 30-40 generations

ago, with a sample size of only 8 individuals, these observations are not conclusive. Our failure to detect drift in these populations could, therefore, also be due to the fact that we have a small sample size and/or relatively recent divergence and the existing methods are not well suited to discern drifts with small sample sizes at this time scale. We are happy to include these results in the main text and the Supplementary Note, if the reviewer considers that to be necessary.

Similarly, in Figure 2D, it appears as though the trend in F_{ST} is simply recapitulating the trend in mixing proportions, with the Khwe having the least Khoisan ancestry and the Ju/'hoansi having the most Khoisan ancestry. Note that differences in mixing proportions violates the assumptions of Weir and Cockerham's F_{ST} .

Response: We thank the reviewer for directing us to this interesting aspect. Though F_{ST} has been applied quite widely to admixed population including Southern African populations (South Eastern Bantu) the complex admixture between the Bantu-speakers and the KS has the potential to bias the results as pointed out by the reviewer. To address this concern we have performed a **local ancestry analysis** to identify the YRI ancestry regions (using PCAdmix) and recomputed F_{ST} scores for these regions only. The results have been summarized in Figure 10b and show an overall concordance in the two studies. We have added the following section in the supplementary note to explain it:

“As all three populations sequenced in our study have some degree of KS admixture, there was a possibility that these admixture rates could have affected the F_{ST} estimates between the study populations and various KS populations. To reduce the effect of admixture in the F_{ST} estimates we employed an approach similar to Gurudasani et al. 2015, in which we identified and masked the non-Niger-Kordofanian (NK) ancestry and computed F_{ST} in the masked datasets. For this the local ancestry detection tool, PCAdmix was used with Ju/'hoansi (as the proxy for the KS), YRI (as the proxy for NK) and CEU (from KGP) as three ancestral populations to identify 20 SNP genomic segments that show >80% in the SEB2 from the Schlebusch et al. 2012 dataset. Based on the local ancestry estimates, the non-NK regions were masked and F_{ST} was calculated for the masked dataset. We added:

“The results summarized in **Supplementary Figure 10a** show the estimates of genetic distance between SOT, XHS, COL and various KS groups to be similar to that observed in Figure 3d, confirming that the estimates are robust to the effects of admixture.”

The authors correctly point out the problem in the f_2 analysis given the unequal sample sizes between the 1000 Genomes data and their own data. I recommend randomly drawing subsets of individuals from the 1000 Genome data.

Response: The observation points out a limitation of the f_2 analysis to detect admixture and rather focuses on identifying the closest relatives. This is illustrated by the absence of discernable second or subsequent ancestry in known admixed populations such as ACB, ASW, MXL, and BEB in the KGP Phase publication.

As, by design, f_2 is supposed to include all samples, as unless we include all samples from a dataset, the definition of f_2 i.e a SNP is observed only once in the dataset, is violated, so bootstrapping or similar analyses were not feasible. To minimize the loss of real f_2 variants due to sample size differences, we therefore performed an overall SNP sharing analysis.

Based on the reviewers suggestion we also performed an analysis of continent-specific SNP sharing. The results summarized in Supplementary Figure 9c were similar to that observed for f2 analysis. Taken together this indicates that the limitation to detect multiple ancestries was limited by the design of the analysis and not the sample sizes.

P. 2: The statement regarding the number of singletons is puzzling. Singletons are always expected, regardless of sample size. Do the authors mean that the number of singletons was not different from the expected value, given the sample size?

Response: We thank the reviewer for pointing out this ambiguity in the statement. We have added a site frequency spectrum based figure (Figure 1f) to demonstrate this and corresponding changes have been made in the main text (Page -2 penultimate paragraph last line) and supplementary notes sections. The sentence has been modified as follows: "A significant proportion of the SNVs identified were singletons. The number of singletons in the SOT and XHS was found to be higher in comparison to the COL and to the number of singletons detected in SFS of randomly selected low-coverage African WGS sets of the same sample size. (Fig.1e)."

We also made the following addition to supplementary note:

"Site Frequency Spectrum (SFS)

As an additional measure to evaluate of quality of sequencing we analyzed the SFS in the three populations and compared them to the SFS observed in the same number of randomly selected individuals from KGP (YRI and ASW, CEU) and AGVP (ZUL). We observed an overall agreement with the SFS in African populations. Moreover, a slightly higher proportion of singletons in SOT and XHS in comparison to the COL as well to random sample sets of same size obtained from various populations from African, Non-African and African-admixed populations was observed (Figure 1f). The variation in SFS between populations is well known and studies have shown African populations to harbour more rare variants compared to non-African populations (Fu et al, 2012, Henn et al. 2015). The relatively lower rate of singletons in the COL (as well as ASW, the other admixed population) can be explained on the basis of this.

While the differences in the SFS from the random African populations compared to SOT and XHS might lead to the hypothesis for an elevated singleton rate due to Khoesan admixture in the SOT and XHS, these differences could as well have been caused by differences in the sequencing depths of the two datasets. Han and colleagues' investigation (Han et al. 2014) on SFS and its relationship to sequencing depth and the variant calling approach employed has shown that estimating genotypes by pooling individuals in a sample set (multisample calling) in low coverage data, as used in KGP Phase-3 and the AGVP studies, results in underestimation of the number of rare variants, which aligns with our observation. However, the slight enrichment of singletons in the SFS of the Zulu in comparison to YRI, both of which are low-coverage datasets, suggests that observed differences in addition to technical differences might also be due to geography/admixture-based differences (Figure 1f). High-coverage sequence datasets of comparable sample sizes from other African populations would enable us the investigation of the observed differences in follow up studies."

In Table 2, what data suggest that any Y haplogroup E has a K-S origin?

Response: The Y haplogroup E has also been found in K-S individuals, but there is no evidence that they are of K-S origin. We have changed this in Table 2 to just indicate that they are of “African origin”.

In Figure 1d, indels is misspelled in the y-axis label.

Response: Corrected, thank you.

REVIEWERS' COMMENTS:

Reviewer #1 (Remarks to the Author):

The original work was exciting, and the revisions made have substantially improved the work addressing many of the reproducibility concerns. My only remaining concern is data access. The URL specified does not work, and I do understand that you are waiting for ethics approval to activate the link. However it is more normal to submit data to a public repository, such as NCBI or ENA which have mechanisms to support the distribution of your exciting data. I would strongly recommend submitting to ENA so that others can benefit from your work.

[Please also review Attachment #1]

Reviewer #2 (Remarks to the Author):

The authors have successfully responded to my concerns and suggestions.

It is concerning that the process of data sharing is still pending IRB approval. The authors indicated that the WGS and Genotyping Array Data will be available through application on the SAHGP website (URL: <http://saghp.sanbi.ac.za/index.php>) but awaiting IRB approval. Is there a possibility that the IRB application will be disapproved? Given the potential importance of this data to the scientific community, it will be optimum if the part to data sharing is more firmly established. It is also not clear what the authors mean by "The data can also be "searched" through the Global Alliance for Genomics and Health "Beacon project".

Attachment #1

Responses to reviewer comments: Reviewer #1 (Remarks to the Author): □ □

(A) Concerns on data production and availability □ □ □ (A1) Data availability: Download details for the bams and vcfs (variant files) should be provided. □ □ **Response:**

WGS and Genotyping Array Data will be available through application on the SAHGP website (URL: <http://saghp.sanbi.ac.za/index.php>). We would be happy to provide a link to our server at the University of the Witwatersrand for the reviewers to access that data for review purposes. We are awaiting ethics approval to activate the link to the data request process. The data can also be "searched" through the Global Alliance for Genomics and Health "Beacon project".

Yes: please provide the link for review as, the specified URL is indeed inactive.

(A2) Consistency of analysis: □ Was a consistent wetlab and informatics procedure used for all samples? Is there any possibility of batch effects that might explain the differences observed? □ □ **Response:**

The DNA was extracted in the same laboratory using a modification of the salting out procedure. The DNA was normalized and sent to the service provider (Illumina Fast Track) as a single batch at the same time and all the data were returned in one batch.

This is great.

(A3) Alignment details are extremely terse: The authors write: "Variants were called according to the GATK best practices for variant calling from cohorts of samples, using the BAM files generated by the Isaac Aligner." This is not reproducible, because the 'best practices' change with new developments. How many raw reads were generated for each sample? Were duplicates

removed (if so how many) - what procedure was followed? What software? What versions of the software? **Response: A table specifying the number of reads and duplicates marked has been added as Supplementary Table 1a. The procedure was added to the online methods and details are available in the supplementary note.**

These additions are good.

Were quality scores recalibrated? Was indel-realignment performed, which should lead to more reliable genotypes and indels? Ideally a table summarising each of these steps, in counts of reads for each sample, should be provided. It would also be useful to know if adapter trimming was performed. If it were not, this could lead to false genotypes to be called. □

Response:

Information about the base quality score recalibration and indel realignment and adaptor trimming was added to online methods and supplementary note. The parameters for the scripts for the variant calling have been added to the supplementary note.

Great.

(A4) Variant filtering: □ There is no discussion of filtering of variants. Surprisingly, the word 'filter' does not appear in the supplementary 1 where one might expect to see this, and in fact appears once and once only in all the supplementaries. This is typically a huge component of variant analysis, and can be very challenging. Filtering is mentioned in Supplementary 4 for PCA analysis which does not affect the raw counts of variants found. Alignments have been generated using a particular aligner (Isaac), and then variants are called using essentially two independent methods (the method used by University of

Pretoria is likely to only generate a subset of the variants from that of the University of Witwatersrand). Taking an intersection of the 3 (well, two really) methods is not adequate to remove false positives. Common filters to use would be mappability filters, tandem repeat filters etc, which help eliminate artifact variants which arise from common misalignment regions. False positives could easily pass through both genotyping methods and would inflate the 'number of unique variants found from these samples'.
Response: Details on filtering of variants has been added to the Online Methods and the Supplementary Notes.

The new supplementary tables and modified Online Methods, along with the snp statistics is great.

Therefore, I would suggest minimally:

(A4.1) a venn diagram of the 3 variant calling methods, showing counts for method specific methods,
Response: The Venn diagrams with counts per sample have been included in Supplementary Figure 1c.

The venn diagram suggest very good consensus between the methods which is encouraging. I would suggest adding numbers to this.

(A4.2) Transition-transversion metrics. The vast majority of new variants according to Figure 1(f) have MAC=1 which is understandable given the low sample count. However, there is no indication of whether these are real. Transition-transversion metrics of the variants, both for the entire set and stratified by MAC would indicate if these are true variants. Ideally, Ts- Tv rates as a function of filters used would be good. I note that the genotype concordance with the Omni chip is reported at 99.5%

however, this tells us nothing about the variants called at sites which are not on the genotyping array. And a PCA analysis does not help, because it is not sensitive to random noise from false positives. □

Response: A plot of the TsTv ratio as a function of minor allele count has been added as Supplementary Figure 1b. All ratios were in the acceptable range of 2 - 2.2. A comment was added to online methods and supplementary text. A table of TsTv ratio per sample stratified by variants in dbSNP_183 and novel SNPs has been added as Supplementary Table 2b.

This is excellent.

(A4.3) specify the specific parameters used for variant calling in the supplementary text (the authors have noted that these are available on request), but it is better to report them as they are not typically verbose. □ Another possibility might be to look at the within-population allele-frequency spectrum. An additional possibility is to do a PSMC analysis of variants; an inflated recent population size would point to inadequate filtering. □

Response: Details were added in the supplementary text regarding - Raw read processing, BQSR, reason for not doing indel realignment, HC parameters, VQSR parameters. We would be happy to add a link to scripts that we can place online if necessary. We have re-written the relevant sections in the Online methods and Supplementary Notes.

We have added an allele frequency spectrum analysis as suggested that shows the number of sites in each MAF bin largely corresponds to the expected values based on random African sample WGS sample sets of same size (and added Figure 1f) and described the observations in the Supplementary Note.

Excellent.

(A4.4) Question: Variants for 1kg with $MAF > 0.01$ were removed, which means that lower frequency variants found in 1kg are ignored. This might also inflate the number of new

variants reported. Could the authors comment on the choice of this threshold and what would happen if it were

lowered? □ □ Response: Thanks for highlighting the confusion.

These variants were not removed from the core data at any point but were removed for specific analyses such as PCA and admixture. This filtering was therefore only performed for the PCA and ADMIXTURE analysis, to reduce the impact of low-frequency and dataset specific variants. We completely agree with the reviewer's concern and would like to confirm that we did not use this filtering for any of the other analyses. This is now clearly documented in the text.

This is great.

(B) Concerns about analysis □ □ □ Given the lack of details about variants, analytical conclusions are worrisome, and might be

artifacts. Some specific concerns follow. □ □

(B1) Criticism of MT handling. The authors note that: "For each sample, reads were first aligned to the GATK reference bundle. Reads mapping to MT were then re-aligned to the Reconstructed Sapiens Reference Sequence (RSRS)⁴³ and genotypes called using GATK Haplotypecaller. The calls were compared to RSRS to create a list of mutations for the sample. These mutations were then compared to each terminal node in Phylotree version 1444 and the terminal node with the most matching parents was chosen as the most likely haplogroup." (p7 Main paper). □ □ This procedure is problematic in several ways: □ i) numts (mitochondrial

sequences homologous to autosomal sequences) are ignored, which leads to read loss and potential bias in the mt coverage. There are 755 known numts, ranging in size from 39bp to almost the whole mt sequence. A better procedure is to either align all reads to the RSRS sequence (which is fairly quick because the MT genome is small), or less preferable, to identify all reads from both the numts and MT genome, and align these to the RSRS. □ ii) MT haplogroup calling here seems to have been done in a manual (and potentially error-prone) way, as I don't believe phylotree provides haplogroup assignment. If I understand the authors correctly - it would be better to use an automated tool such as: Haplogrep, mthap, or Haplofind.

Response: As suggested by the reviewer we have used Haplogrep2 to repeat the mtDNA haplogroup analysis and essentially identified the same haplogroups, with one exception. This, however, did not affect the nature of the origins of the mtDNA lineages and the main conclusion stays the same. The results, methods and references were updated accordingly.

Great.

(B2) f2 variant analysis is a fairly recent technique and should be referenced, eg: Mathieson I, McVean G (2014) Demography and the Age of Rare Variants. PLoS Genet 10(8): e1004528. doi:10.1371/journal.pgen.1004528 or similar. □ □ Could the authors comment on Figure 3 of the main paper: in the f2 analysis, we should expect that the COL population which is a known mixture of Indian and other populations, should share more f2 variants with 1kg south asian populations such as GIH and BEB, yet it seems that there are actually fewer compared with the SOT sample. Does this seem surprising? □

Response: As suggested by the reviewer we also expected to see

some of the other ancestries in the COL to be captured in the f_2 analysis. Therefore, while describing this in the supplementary notes, we have included a few lines stating this and speculated on the possible reasons for such an observation (Supplementary Notes Page 9 last paragraph). We also performed an overall SNP sharing analysis to minimize the loss of real f_2 variants due to sample size differences. The results are summarized in Supplementary Figure 9, and were similar to that observed for f_2 analysis.

We believe that the problem is that f_2 analysis has not been designed to detect admixture as seen in the COL. A scenario where a variant is shared between a population and a distant offshore relative (due an historical admixture event) but not with any neighboring present day population from the same geographic region, might be problematic. For example, in the Phase 3 KGP analysis we do not see any significant second or subsequent component for known admixed populations such as ACB ASW, MXL, and BEB. Therefore, the major ancestry in the Southern African groups, as expected, was the East African Bantu which makes sense in terms of known models of Bantu migration as well as history.

An intuitive alternative to this appeared to be the study of SNPs that are shared by populations in a continent and its offshore relatives. We have provided an additional analysis aimed at studying continent-specific SNP sharing rather than population-specific SNP sharing (as implemented in f_2). The results from this analysis, integrated into Supplementary Figure 9, clearly shows that analysis of continent-specific SNP sharing yields similar results. A line has been added to the main text and a paragraph has been added to the supplementary note to describe this analysis.

This clarifies the f_2 analysis, and the additional analysis is valuable.

(B3) Details missing in F_{ST} analysis:

In the F_{ST} analysis (Section 9, supplementaries), an important analysis to measure the difference between populations, it would be useful to see details. For examples, the authors write: "For this, a merged dataset constituting of data from the SAHGP, AGVP and Sclebusch et al. 2012 studies 4,7 was generated. ". However, there's no mention of the number of snps that result, the number that are removed and filtering (based on say, missingness) is not mentioned. **Response: We thank the reviewer for pointing out this gap in the methods section. A paragraph has been added to the online methods section and additional references were added (Page 9 first paragraph) (see below).**

"For merging the above-mentioned datasets genotyped on Omni 2.5M SNP chip, SNPs that has been successfully genotyped in all the three studies were identified. The genotype data corresponding to these SNPs were extracted from each dataset and merged using PLINK1.9. The dataset thus generated contained 1,028,376 SNPs and 3,108 individuals. After removing SNPs that show discordance in alleles, filtering based on both individual level missingness of >0.05 , SNP level missingness of >0.05 , and HWE cut of $p < 0.0000001$, retained 1,026,664 SNPs and 3,108 individuals for F_{ST} analysis."

Great.

(B4) Regions of divergence between the Bantu speaking groups. This was interesting, and section 6 was

good? Response: Thanks.

(B5) Supplementary Table1: Sequencing Coverage observed in the 24 samples. □ □ The XHS samples have consistently lower coverage than almost all the other samples - do the authors believe this is due to reference bias, given the hg19 reference is a mixture of other varying groups which do not include XHS. □

Response: We appreciate the reviewer's careful scrutiny of the table. Statistical tests

support this observation, to some extent. The t-test based P-value for the differences between mean coverage in COL and XHS is <0.001 . However, the P-value for differences in COL and SOT is not significant (0.8). Moreover, the P-value of difference between SOT and XHO is also not significant, even if we remove the SOT individual with very low coverage. Therefore, while there is a slight trend in the data and reference bias seems a logical source for the observed differences in coverage, the evidence in the data is not strong enough to form the basis of any hypothesis. We have added a paragraph in the supplementary note stating this (as follows).

“Although we observed that the overall coverage in the XHO samples was lower than in the other two populations a t-test based evaluation showed the differences to be statistically significant only in the XHS-COL, but not in the other two comparisons (SOT-COL and XHS- COL). Therefore, the observed differences do not appear to be related to demographic history or ancestry.”

This still seems a little surprising, ; some reference bias seems inevitable here, but it probably does not affect any conclusions, and it is good that this has been followed up statistically.

Minor typo: □ "Supplementary Table 13: P-valuesr for differences in total ROH lengths of individuals from all 49 populations" (in the word 'P-values'). □ **Response: Corrected**

Reviewer #2 (Remarks to the Author):

Strengths of this manuscript are variant discovery in understudied ancestral populations and local genomic capacity development in a low resource environment. There are however some weaknesses the most significant being the small number of individuals sequenced and the methodologies used to analyze generated sequence data.

Response:

The methods are now described in much more detail and in line with the recommendations from reviewers 1 and 2.

The analysis basically reduced the sequence data to genotype by not implementing the latest analytic strategies for the analysis of sequence data – for example, the Pairwise Sequentially Markovian Coalescent model or the Multiple Sequentially Markovian Coalescent model for the analysis of whole genome sequences to infer human population history.

Response: We acknowledge this limitation. The major factor that made us reduce the data for some of the analyses (PCA and STRUCTURE) to chip data, as the reviewer has pointed out, is the unavailability of population level WGS from southern African hunter-gatherers (the whole genome data in Lachance et.al. 2012 were from eastern and central Africa. The southern African hunter-gatherer data in Kim et al. 2014 included only 2 individuals per group from two hunter-gatherer groups and therefore were insufficient for most analyses). Population level datasets from southern African hunter-gathers will enable future researchers to perform many of these studies at the WGS level.

We also completely agree about the MSMC analysis. There were a couple of issues, which did not allow us to include the analysis in the first draft. Firstly, although being widely used, the tool is quite complex in terms of integration of data from independent studies and also computationally intensive. Moreover, the high Khoesan admixture in the SAHGP populations and its effect on MSMC based estimates needed evaluation. We have set up

collaboration with a research group with expertise in this area and initiated the analysis. We hope to be able to include this analysis if given an opportunity to resubmit.

In addition, it is not immediately apparent what novel population genetics insights were provided by the sequencing of these small number of individuals that we do not already know. □ □ **Response:** The novelty is that we found significant differentiation between the SO and XHO who have only been geographically separated for ~1200 years. We also provide analyses and discussion on the potential drivers for this divergence. In addition, we have shown that despite the fact that there is now a lot more WGS data available, we still detected ~0.8M novel SNVs. We have shown that these variants are not equally distributed throughout the genome. Moreover, this will be the first **high-coverage WGS data** from Southern African Bantu-speakers and also the highly admixed coloured population and would provide future researchers an important dataset for more detailed analysis of admixture and local ancestry as the hunter-gatherer WGS data from this region becomes available.

In order to provide context for whole genome sequencing, the Introduction should refer to SC Schuster et al. (2010 Nature 463:943-947) and HL Kim et al. (2014 Nat Commun 5:5692). The published literature contains papers not cited in this manuscript that address ancestry in relevant groups of South Africans, e.g., E

de Wit et al. (2010 Hum Genet 128:145-153), D Shriner et al. (2014 Sci Rep 4:6055), and ER Chimusa et al. (2015 PLOS Genet 11:e1005052). □ **Response: Reference to these studies has been included and the studies have been given context.**

Looking at Figure 2C, it appears that the SOT and XHS are essentially two-way mixtures of ancestries shared among Khoisan (shown in light blue) and Southeastern Bantu (shown in red) speakers, although there are no error bars. Thus, it appears that the ancestral difference between the SOT and XHS is not qualitative but quantitative. It is not clear whether differentiation between the Nguni and Sotho-Tswana speakers reflects (1) different mixture proportions, with the SOT having a slightly higher percentage of Khoisan ancestry, or (2) divergence due to random genetic drift from a common ancestor.

Response: As suggested by the reviewer, the differences between the SOT and XHS seem largely quantitative and suggest possibility (1) as the source of the difference between the two populations. However, we agree that it is also important to investigate whether the alternative (2) also plays some role in the observed differentiation of these populations. Site frequency spectrum comparisons have been shown to correspond to, and vary according to, genetic drift at least between various continental populations (Keinan et al. 2007). The high similarity in the site frequency spectrum of SOT and XHS suggests that there has been little drift after separation in these populations. Similarly, we could not find any notable difference in the level of heterozygosity, heterozygosity/homozygosity ratios, IBD or inbreeding coefficient, which has also been suggested to accompany genetic drift. To explore the role of genetic drift further, we also performed an out-group f_3 analysis, which again showed little evidence of genetic drift.

As we were not able to find a method that has been shown to be able to discern the difference between drift and admixture in populations diverged around 30-40 generations

ago, with a sample size of only 8 individuals, these observations are not conclusive. Our failure to detect drift in these populations could, therefore, also be due to the fact that we have a small sample size and/or relatively recent divergence and the existing methods are not well suited to discern drifts with small sample sizes at this time scale. We are happy to include these results in the main text and the Supplementary Note, if the reviewer considers that to be necessary.

Similarly, in Figure 2D, it appears as though the trend in F_{ST} is simply recapitulating the trend in mixing proportions, with the Khwe having the least Khoisan ancestry and the Ju/'hoansi having the most Khoisan ancestry. Note that differences in mixing proportions violates the assumptions of Weir and Cockerham's F_{ST} . □

Response: We thank the reviewer for directing us to this interesting aspect. Though F_{ST} has been applied quite widely to admixed population including Southern African populations (South Eastern Bantu) the complex admixture between the Bantu-speakers and the KS has the potential to bias the results as pointed out by the reviewer. To address this concern we have performed a **local ancestry analysis** to identify the YRI ancestry regions (using PCAdmix) and recomputed F_{ST} scores for these regions only. The results have been summarized in Figure 10b and show an overall concordance in the two studies. We have added the following section in the supplementary note to explain it:

“As all three populations sequenced in our study have some degree of KS admixture, there was a possibility that these

admixture rates could have affected the F_{ST} estimates between the study populations and various KS populations. To reduce the effect of admixture in the F_{ST} estimates we employed an approach similar to Gurudasani et al. 2015, in which we identified and masked the non-Niger-Kordofanian (NK) ancestry and computed F_{ST} in the masked datasets. For this the local ancestry detection tool, PCAdmix was used with Ju/'hoansi (as the proxy for the KS), YRI (as the proxy for NK) and CEU (from KGP) as three ancestral populations to identify 20 SNP genomic segments that show >80% in the SEB2 from the Schlebusch et al. 2012 dataset. Based on the local ancestry estimates, the non-NK regions were masked and F_{ST} was calculated for the masked dataset. We added:

“The results summarized in **Supplementary Figure 10a** show the estimates of genetic distance between SOT, XHS, COL and various KS groups to be similar to that observed in Figure 3d, confirming that the estimates are robust to the effects of admixture.”

The authors correctly point out the problem in the f_2 analysis given the unequal sample sizes between the 1000 Genomes data and their own data. I recommend randomly drawing subsets of individuals from the 1000 Genome data. □ □ **Response:** The observation points out a limitation of the f_2 analysis to detect admixture and rather focuses on identifying the closest relatives. This is illustrated by the absence of discernable second or subsequent ancestry in known admixed populations such as ACB, ASW, MXL, and BEB in the KGP Phase publication.

As, by design, f_2 is supposed to include all samples, as unless we include all samples from a dataset, the definition of f_2 i.e a SNP is observed only once in the dataset, is violated, so bootstrapping or similar analyses were not feasible. To minimize the loss of real f_2 variants due to sample size differences, we therefore performed an overall SNP sharing analysis.

Based on the reviewers suggestion we also performed an analysis of continent-specific SNP sharing. The results summarized in Supplementary Figure 9c were similar to that observed for f2 analysis. Taken together this indicates that the limitation to detect multiple ancestries was limited by the design of the analysis and not the sample sizes.

P. 2: The statement regarding the number of singletons is puzzling. Singletons are always expected, regardless of sample size. Do the authors mean that the number of singletons was not different from the expected value, given the sample

size? □ □ Response: We thank the reviewer for pointing out this ambiguity in the statement. We have added a site frequency spectrum based figure (Figure 1f) to demonstrate this and corresponding changes have been made in the main text (Page -2 penultimate paragraph last line) and supplementary notes sections. The sentence has been modified as follows:

“A significant proportion of the SNVs identified were singletons. The number of singletons in the SOT and XHS was found to be higher in comparison to the COL and to the number of singletons detected in SFS of randomly selected low-coverage African WGS sets of the same sample size. (Fig.1e).”

We also made the following addition to supplementary note:

“Site Frequency Spectrum (SFS)

As an additional measure to evaluate of quality of sequencing we analyzed the SFS in the three populations and compared them to the SFS observed in the same number of randomly selected individuals from KGP (YRI and ASW, CEU) and AGVP (ZUL). We observed an overall agreement with the SFS in African populations. Moreover, a slightly higher proportion of singletons in SOT and XHS in comparison to the COL as well to random

sample sets of same size obtained from various populations from African, Non-African and African- admixed populations was observed (Figure 1f). The variation in SFS between populations is well known and studies have shown African populations to harbour more rare variants compared to non-African populations (Fu et al, 2012, Henn et al. 2015). The relatively lower rate of singletons in the COL (as well as ASW, the other admixed population) can be explained on the basis of this.

While the differences in the SFS from the random African populations compared to SOT and XHS might lead to the hypothesis for an elevated singleton rate due to Khoesan admixture in the SOT and XHS, these differences could as well have been caused by differences in the sequencing depths of the two datasets. Han and colleagues' investigation (Han et al. 2014) on SFS and its relationship to sequencing depth and the variant calling approach employed has shown that estimating genotypes by pooling individuals in a sample set (multisample calling) in low coverage data, as used in KGP Phase-3 and the AGVP studies, results in underestimation of the number of rare variants, which aligns with our observation. However, the slight enrichment of singletons in the SFS of the Zulu in comparison to YRI, both of which are low-coverage datasets, suggests that observed differences in addition to technical differences might also be due to geography/admixture-based differences (Figure 1f). High-coverage sequence datasets of comparable sample sizes from other African populations would enable us the investigation of the observed differences in follow up studies."

In Table 2, what data suggest that any Y haplogroup E has a K-S origin? Response: The Y haplogroup E has also been found in K-S individuals, but there is no evidence that they are of K-S origin. We have changed this in Table 2 to just indicate that they

are of “African origin”. □

In Figure 1d, indels is misspelled in the y-axis label. □ **Response:**
Corrected, thank you.